# Slik sculpts the plasma membrane into cytonemes to control cell-cell communication

Basile Rambaud[1,2,7], Mathieu Joseph[2,3,7], Feng-Ching Tsai[4], Camille De Jamblinne[1,2], Regina Strakhova[1,2], Emmanuelle Del Guidice[1,2], Renata Sabelli[1,2], Matthew J Smith[1,2,5], Patricia Bassereau [iD] [4], David R Hipfner [iD] [2,3,6 ✉] & Sébastien Carréno [iD] [1,2,5 ✉]

## Abstract

**Cytonemes are signaling filopodia that facilitate long-range cell–cell communication by forming synapses between cells. Initially discovered in *Drosophila* for transporting morphogens during embryogenesis, they have since been identified in mammalian cells and implicated in carcinogenesis. Despite their importance, mechanisms controlling cytoneme biogenesis remain elusive. Here, we demonstrate that the Ser/Thr kinase Slik drives remote cell proliferation by promoting cytoneme formation. This function depends on the coiled-coil domain of Slik (SlikCCD), which directly sculpts membranes into tubules. Importantly, Slik plays opposing roles in cytoneme biogenesis: its membrane-sculpting activity promotes cytoneme formation, but this is counteracted by its kinase activity, which enhances actin association with the plasma membrane via Moesin phosphorylation. In vivo, SlikCCD enhances cytoneme formation in one epithelial layer of the wing disc to promote cell proliferation in an adjacent layer. Finally, this function relies on the STRIPAK complex, which controls cytoneme formation and governs proliferation at a distance by regulating Slik association with the plasma membrane. Our study unveils an unexpected structural role of a kinase in sculpting membranes, crucial for cytoneme-mediated control of cell proliferation.**

**Keywords** Slik Kinase; Cytoneme Biogenesis; Membrane Sculpting; STRIPAK Complex; ERM Protein
**Subject Categories** Cell Adhesion, Polarity & Cytoskeleton; Development; Membranes & Trafficking

## Introduction

Pioneering work on morphogens established that cells located in 'organizing centers' secrete ligands to control the fate of distant cells in a 'cell non-autonomous' signaling process (Ng et al, 1999).

These morphogens were initially thought to be secreted directly into the extracellular space, where their passive diffusion dictates concentration-dependent responses on distant cells (Rogers and Schier, 2011). Yet, this mechanism cannot fully account for some long-range signaling needed during embryogenesis. An alternative mechanism that also creates robust morphogen gradients was more recently identified (Kornberg and Roy, 2014). It involves long actin-based signaling filopodia called cytonemes that synapse between signal-sending and -receiving cells (Daly et al, 2022). Cytonemes were discovered in Drosophila wing imaginal discs (Ramirez-Weber and Kornberg, 1999), composed of two distinct but continuous layers of epithelial cells. Morphogens and their receptors traffic within or along cytonemes in vesicles (Bischoff et al, 2013; Callejo et al, 2011; Chen et al, 2017; Hsiung et al, 2005; Huang and Kornberg, 2015; Peng et al, 2012; Roy et al, 2011). Cytonemes have also been identified in vertebrates and mammals (Hall et al, 2024; Hall et al, 2021; Sanders et al, 2013; Stanganello et al, 2015; Zhang et al, 2024). They have drawn increased interest since cancer cells use cytonemes and related structures like tunneling nanotubes to communicate with each other or with stromal cells (Fereres et al, 2019; Mattes et al, 2018; Pinto et al, 2020; Rogers et al, 2023; Routledge et al, 2022). Relatively few of the proteins that participate in cytoneme formation and function have been identified (Bischoff et al, 2013; Hall et al, 2024; Routledge et al, 2022; Roy et al, 2011), and our understanding of the mechanisms controlling their biogenesis is limited. Cultured Drosophila S2 cells form cytonemes, transducing cell–cell Hedgehog signaling, making them a potent model for characterizing cytoneme biogenesis (Bodeen et al, 2017).

We previously demonstrated that the Ser/Thr kinase Slik plays an important role in governing cell–cell communication between epithelia layers in Drosophila. Expression of Slik in cells of the "disc proper" (DP) epithelium of the wing disc promotes proliferation in the adjacent peripodial membrane (PerM) epithelium, separated by a lumen (Hipfner and Cohen, 2003; Panneton et al, 2015). Slik is a GCK-V subfamily Ste20 kinase conserved from fly to human. Slik has three domains: the N-terminal kinase domain, a central non-conserved region, and a conserved coiled-coil C-terminal domain (SlikCCD) (Hipfner and Cohen, 2003). Interestingly, we also reported

[1]Institute for Research in Immunology and Cancer (IRIC), Université de Montréal, Montreal, Quebec H3C 3J7, Canada. [2]Programmes de biologie moléculaire, Université de Montréal, Montreal, Quebec H3C 3J7, Canada. [3]Institut de recherches cliniques de Montréal (IRCM), Montreal, Quebec H2W 1R7, Canada. [4]Institut Curie, Université PSL, Sorbonne Université, CNRS UMR168, Physics of Cells and Cancer, 75005 Paris, France. [5]Département de Pathologie et Biologie cellulaire, Université de Montréal, Montreal, Quebec H3C 3J7, Canada. [6]Département de Médecine, Université de Montréal, Montreal, Quebec H3C 3J7, Canada. [7]These authors contributed equally: Basile Rambaud, Mathieu Joseph. ✉E-mail: david.hipfner@ircm.qc.ca; sebastien.carreno@umontreal.ca

that Slik function in cell–cell communication does not require Slik kinase activity (Panneton et al, 2015). Yet, how Slik controls proliferation at a distance remains unknown.

Here, by combining a series of complementary approaches in vitro, in S2 cells and in vivo in wing discs, we show that Slik acts as a novel regulator of cytoneme biogenesis to control proliferation at a distance. We identified a pivotal role of the C-terminal coiled-coil domain of Slik in promoting cytoneme biogenesis. Like wild-type Slik or a mutant without kinase activity, expression of Slik$^{CCD}$ alone increases cytoneme number and length in S2 cells and induces over-proliferation in vivo. Importantly, we discovered that Slik$^{CCD}$ nucleates and elongates cytonemes by directly reshaping membranes, similar to Bin/amphiphysin/Rvs (BAR) membrane-sculpting domains (Simunovic et al, 2019). We also present evidence that Slik may exert paradoxical activities on cytoneme biogenesis: whereas its CCD promotes cytoneme biogenesis, Slik kinase domain phosphorylates and activates Moesin, the only ERM family protein in the fly (Carreno et al, 2008; Hipfner et al, 2004; Kunda et al, 2008; Polesello et al, 2002), to increase cortical rigidity and counteract cytoneme formation. Finally, we showed that the Drosophila Striatin-interacting phosphatase and kinase (dSTRIPAK) complex controls cytoneme biogenesis and communication at a distance by regulating the association of Slik with the plasma membrane.

# Results

## Slik alters apical membrane morphology and promotes formation of apical cytonemes in wing discs

While investigating the role of dSTRIPAK in regulating the association of Slik with the plasma membrane (De Jamblinne et al, 2020), we noticed that Slik overexpression in cells of the DP layer of the imaginal disc promotes the formation of supra-apical protrusions into the disc lumen. These protrusions were rich in F-actin which formed a second layer atop of the terminal web (Fig. 1A). The apical polarity determinant Crumbs (Crb) localized to these supra-apical protrusions suggesting they represent an expansion of the DP apical membrane (Fig. 1B). Consistent with Slik affecting the apical membrane, immunofluorescence staining of wild-type wing disc cryosections revealed that a fraction of endogenous Slik protein is enriched in the apical-most free membrane region, above the level of adherens junctions, where it co-localized with phospho-Moesin (Fig. EV1).

To characterize these protrusions, we performed live imaging on wing discs expressing a membrane-targeted GFP bearing the Src myristylation sequence (srcEGFP) (Kaltschmidt et al, 2000). In control discs, srcEGFP labeled sparse fine apical membrane protrusions that projected into the disc lumen (Fig. 1C). Upon overexpression of Slik, we observed a significant increase in the number of these protrusions, which often exceeded 4 μm in length (Fig. 1C). We also observed the frequent appearance of larger membrane blebs connected by a stalk to the apical DP cell membrane, which expanded at their tips to fill the lumen and contact the PerM (Fig. 1C, bottom). These were never observed in control discs. When a functional GFP-tagged version of the kinase (Slik-GFP) (Roubinet et al, 2011) was expressed in DP cells, we observed that Slik was strongly enriched at the apical membrane

and localized to the fine protrusions and membrane blebs in live discs (Fig. 1D). Many of these structures were dynamic, showing ruffling and rapid extension/retraction, whereas others appeared more stable (Fig. 1E and Movie EV1).

We used split GFP complementation to test if these protrusions connect DP with PerM cells, as cytonemes would do in order to transport proliferative signals across the lumen. We expressed CD4 tagged with GFP1-10 in the DP cells using the *nubbin-GAL4* driver, while the other GFP fragment (GFP11) fused with CD4 was expressed in PerM cells using a periopodial-specific LexA driver strain (*PerM-LexA*) (Fig. 1F) (see Methods). In control discs, we observed a low but detectable level of GFP complementation, mainly in the disc lumen, indicating some DP-PerM contacts under normal conditions (Fig. 1G, center). Expression of Slik in the DP markedly increased the contact across the lumen with the PerM (Fig. 1G, right). Together these results suggest a model in which Slik expression remodels the apical membrane in DP cells, promoting the formation of apical cytonemes and blebs that transmit a proliferative signal to the PerM cells.

## Slik promotes formation of cytoneme-like protrusions in cultured cells

To obtain deeper insights into how Slik controls cytoneme biogenesis, we turned to cultured S2 cells, derived from Drosophila embryos (Schneider, 1972), which form functional cytonemes (Bodeen et al, 2017). As we observed in wing discs, Slik-GFP localized to actin-rich protrusions, as did the endogenous kinase (Fig. 2A,B). Notably, a substantial portion of these protrusions displayed cytoneme characteristics, distinct from classical filopodia; they did not adhere to the substratum and emerged from the upper region of the cellular body, as illustrated by the 3D cell reconstruction (Fig. 2A). We then employed correlative light scanning electron microscopy (CLSEM) to achieve high-resolution imaging, allowing precise measurement of protrusion diameters while correlating fluorescent signals with ultrastructural details. Using CLSEM, we confirmed that the diameter of Slik-GFP and GFP protrusions were consistent with the diameter of cytonemes (Wood et al, 2021), averaging 172 nm and 181 nm, respectively (Fig. 2C,D). We then investigated whether these protrusions could facilitate the delivery of morphogens to neighboring cells. To test this, we used an established model of co-culture (Bodeen et al, 2017) in which one population of S2 donor cells exogenously express the Hedgehog (Hh) morphogen, while a population of receiving cells express luciferase under the control of the Hh-responsive *patched* (*ptc*) promoter (Fig. 2E). Supporting the role of Slik in promoting formation of bona fide cytonemes, as it does in wing discs, co-expression of Slik-GFP with Hh in donor cells significantly increased *ptc* reporter activity in the receiving cells when compared to donor cells co-expressing Hh and GFP alone (Fig. 2F). Thus, Slik-dependent projections exhibit morphological and functional characteristics consistent with in vivo cytonemes. Although no definitive markers have yet been identified to distinguish cytonemes from non-signaling filopodia, we refer to these structures in S2 cells as cytonemes for clarity and consistency, based on their aforementioned characteristics.

We then further characterized these Slik-dependent cytonemes. Given the delicate nature of cytonemes, which are not entirely preserved using common fixation methods (Bodeen et al, 2017), we

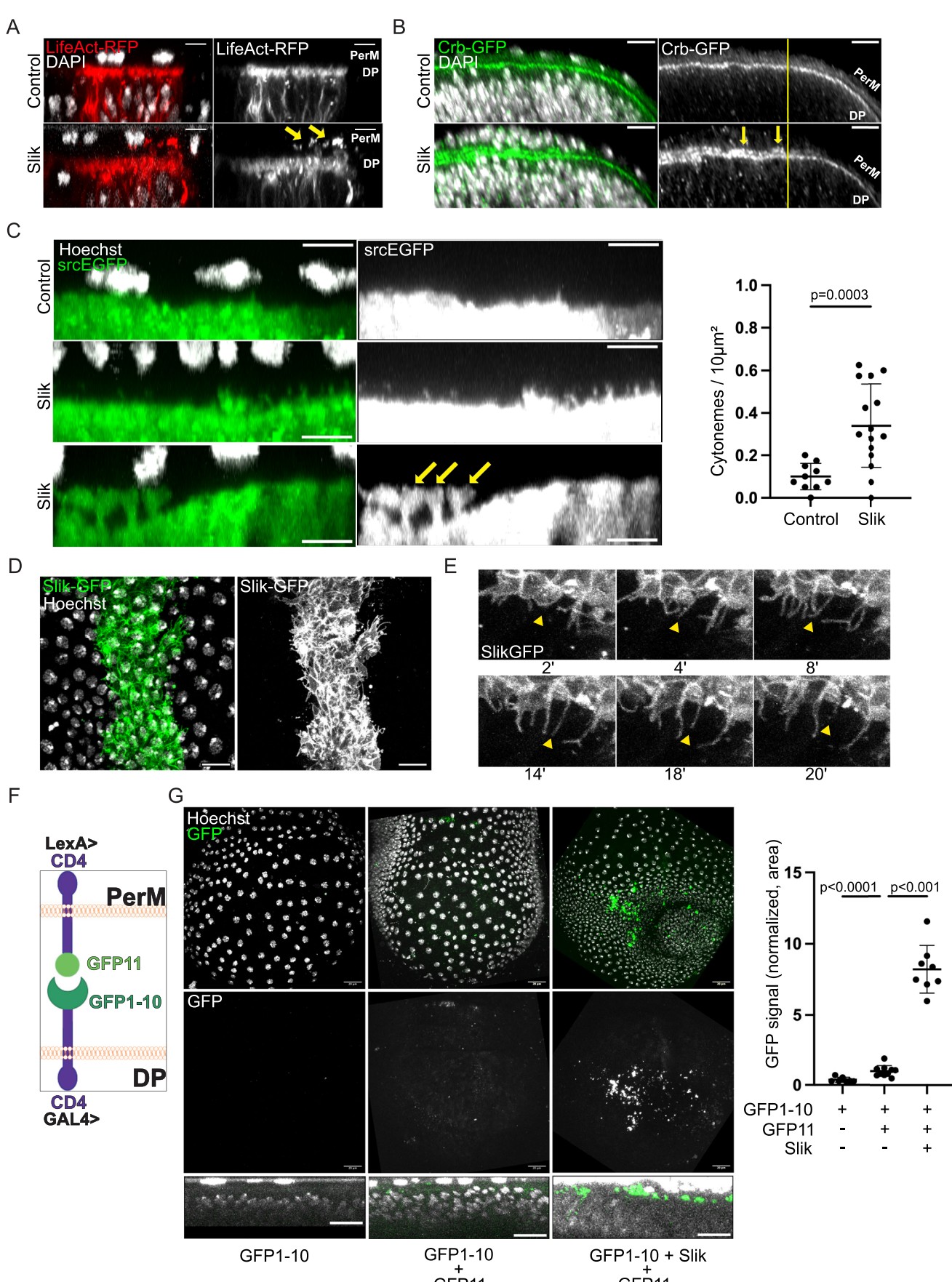

**Figure 1. Slik promotes biogenesis of cytonemes in wing discs.**

(A) Maximum intensity (MAX) projections of confocal microscope XZ images displaying merged channels of fixed wing discs expressing LifeAct-RFP (Red), alone (top panel) or co-expressed with Slik (bottom panel) in a central stripe of DP cells under the control of the *ptcGAL4* driver. DAPI staining reveals nuclei (White). Arrows highlight actin-rich protrusions extending from the DP apical surface into the disc lumen. Scale bars = 5 μm. (B) MAX projection of XZ images showing merged channels of fixed wing discs ubiquitously expressing Crb-GFP (Green), without (top panel) or co-expressed with Slik (bottom panel) in the dorsal compartment using *apGAL4*. DAPI staining visualizes nuclei (White). Yellow lines indicate location of dorsal-ventral boundary, dorsal compartment is to the left. Arrows point to Crb accumulation within the disc lumen. Scale bars = 5 μm. (C) MAX projections of XZ images from live wing discs expressing the membrane marker srcEGFP (Green), alone (top panel) or together with Slik (middle and bottom panels). The bottom image shows membrane blebs. Hoechst staining marks nuclei (White). Graph represents the quantification of the number of fine projections extending from the apical DP surface per unit area. Analysis was performed on 10 (Control) or 15 (Slik) discs. Arrows indicate apical membrane blebs from DP cells making contact with PerM cells. Scale bars = 5 μm. (D) XY image presenting merged channels of a live wing disc expressing Slik-GFP in a central stripe of DP cells (Green). The image is a Z-projection of optical sections covering the peripodial membrane (PerM) (nuclei stained with Hoechst, White) and the apical DP surface. Scale bar = 10 μm. (E) Confocal microscopy timelapse sequence depicting Slik-GFP expression in a wing disc (White). Arrowheads indicate a dynamic cytoneme that extends and then retracts. Scale bar = 5 μm. (F) Schematic representing the assay for split-GFP complementation across the disc lumen. (G) Confocal images showing merged channels of live wing discs expressing CD4-spGFP1-10 in the DP layer alone (left), together with CD4-spGFP11 in the PerM (middle), or with both CD4-spGFP11 in the PerM and Slik in the DP (right). The top row displays MAX Z-projections, while the bottom row shows MAX projections of XZ images. Graph represents the percentage of pixels above threshold intensity per disc for each condition. Analysis was performed on (from left to right) 7, 8, and 10 discs. Scale bar = 20 μm. *P*-values were calculated using unpaired t-test with Welch's correction (C) and unpaired one-way Welch ANOVA with Dunnett T3 multiple comparison (G). Error bars indicate mean ± s.d. Source data are available online for this figure.

quantified the number and length of cytonemes by 3D live-cell imaging. To distinguish overlapping protrusions within cells, we analyzed depth color-coded Z-stack images of maximum intensity projections, allowing us to measure cytoneme signals from planes located 1 μm above the substratum. This allowed accurate measurement of cytonemes that did not contact the substratum (Fig. 2G) (see Methods). Using this approach, we observed that Slik-GFP overexpression significantly increased the number of cytonemes when compared to cells overexpressing GFP (Fig. 2H), a cytoplasmic marker previously used to visualize endogenous cytonemes in S2 cells (Bodeen et al, 2017). Confirming that Slik plays important roles in promoting cytoneme biogenesis, its dsRNA depletion reduced their number (Fig. 2H). Yet, while Slik overexpression increased the length of cytonemes, its depletion did not result in their shortening; instead cytonemes were longer than in control cells (Fig. 2H). This indicates that while Slik promotes the formation of cytonemes, this kinase also plays a complex role in regulating their elongation.

## Moesin counteracts cytoneme biogenesis

Slik activates Moesin, the sole Drosophila ERM member, by phosphorylating its regulatory Threonine[559] (Hipfner et al, 2004). Then Moesin links actin filaments to the plasma membrane, increasing cortical rigidity to control cell shape transformations (Carreno et al, 2008; Kunda et al, 2008; Leguay et al, 2021; Leguay et al, 2022; Roubinet et al, 2011; Solinet et al, 2013). We previously reported that Slik overexpression in S2 cells and in wing discs increased Moesin phosphorylation and activity (Panneton et al, 2015). To investigate whether Slik controls cytoneme biogenesis by activating Moesin, we depleted this ERM using well-characterized *moesin* dsRNA (Carreno et al, 2008). Moesin depletion slightly increased the number of cytonemes in control cells, but did not alter their number in cells overexpressing Slik, indicating that Slik promotes formation of cytonemes independently of Moesin (Fig. 3A). Yet, we found that Moesin negatively regulates cytoneme elongation since its depletion lengthened cytonemes in either GFP or Slik overexpressing cells (Fig. 3A). ERMs regulate the rigidity of the cortex by anchoring the actin cytoskeleton to the plasma membrane (Faure et al, 2004; Kunda et al, 2008). Interestingly,

detachment of the actin cortex following ERM dephosphorylation was shown to precede and to be necessary and sufficient for the initiation of membrane protrusion formation in cells (Welf et al, 2020). We thus hypothesized that Moesin may control cytoneme biogenesis by fine-tuning association of actin with the plasma membrane: strengthening this association increases cortical rigidity and would negatively affect cytoneme biogenesis while detaching actin from the plasma membrane would facilitate cytoneme formation and elongation. To test this hypothesis, we first increased cortical rigidity from the extracellular side of the plasma membrane, independently of Moesin activity, by suspending cells in medium containing the tetravalent lectin, Concanavalin A (Kunda et al, 2008). Consistent with cortical rigidity counteracting cytoneme biogenesis, Concanavalin A treatment reduced both the number and length of cytonemes in control cells (Fig. 3B). We then manipulated actin association with the plasma membrane by expressing phospho-mutants of Moesin that affect its activity (Carreno et al, 2008; Kunda et al, 2008): Moesin[T559D], the Moesin phospho-mimetic, increases cortical rigidity by enhancing actin association with the plasma membrane while Moesin[T559A], its non-phosphorylatable counterpart, relaxes the cortex by promoting actin detachment (Kunda et al, 2008). Expression of Moesin[T559D] decreased both cytoneme number and length, while expression of Moesin[T559A] increased their number per cells (Fig. 3C), suggesting that Slik exerts two paradoxical activities on cytoneme biogenesis. By activating Moesin, Slik increases association of actin with the plasma membrane and counteracts cytoneme biogenesis. However, Slik also promotes cytoneme formation and elongation by an unknown mechanism and independently of Moesin.

The antagonistic roles of Slik in cytoneme formation were also evident in vivo: *slik* depletion in DP cells unexpectedly resulted in an increase in apical protrusions (Fig. EV2). However, unlike the effect of Slik overexpression, these protrusions did not induce non-autonomous proliferation of PerM cells (Fig. EV2). Since Slik is the principal kinase responsible for phosphorylating and activating Moesin in the wing disc (Hipfner et al, 2004), we propose that actin detachment and the resulting reduction in apical membrane stiffness in Slik-depleted DP cells is permissive for the formation of apical protrusions by other mechanisms. Nonetheless, the protrusions formed in the absence of Slik lack pro-proliferative signaling function.

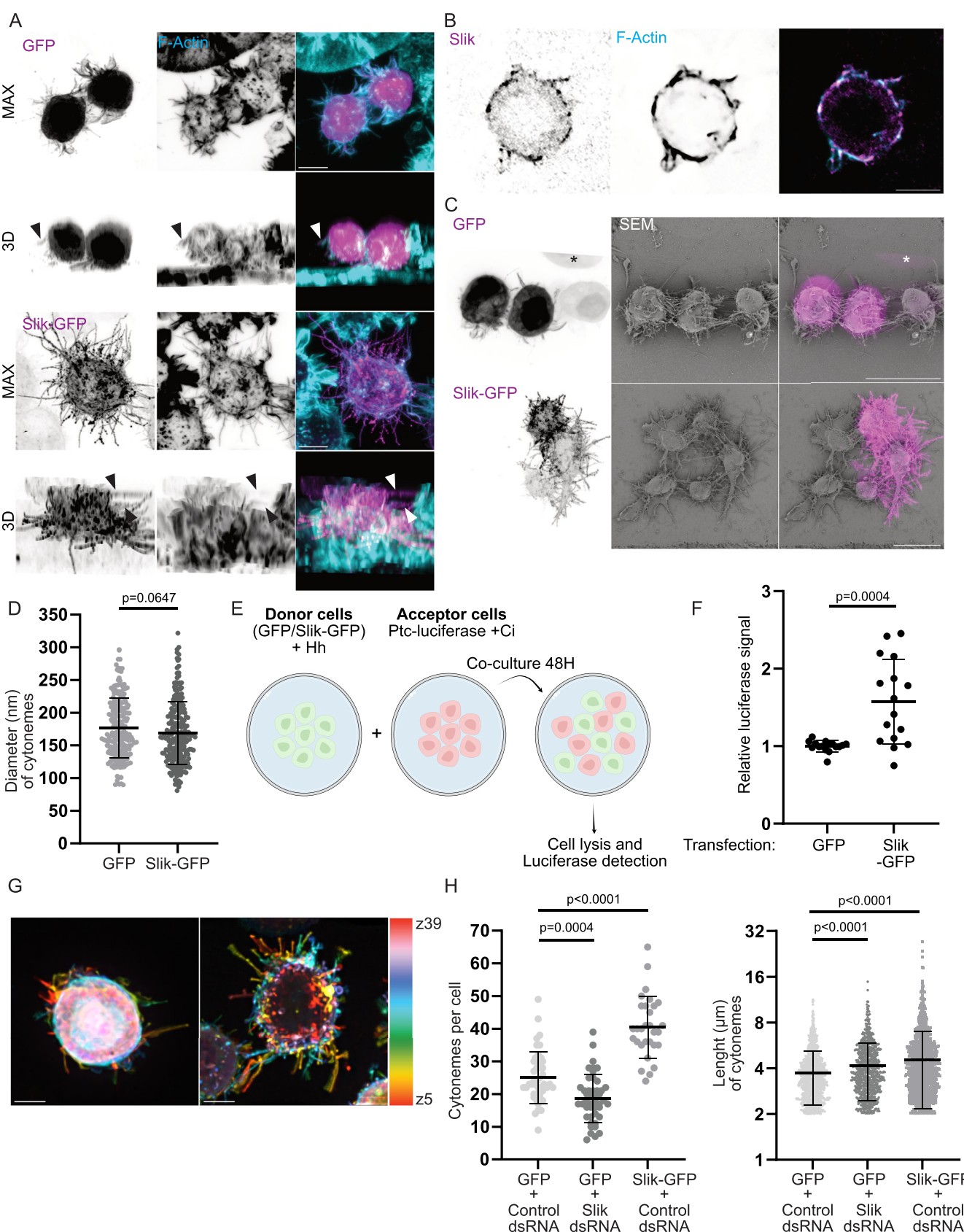

**Figure 2. Slik promotes biogenesis of cytoneme-like protrusions in S2 cells.**

(A) Confocal microscopy images showing merged channels of S2 cells expressing GFP (top, Magenta) or Slik-GFP (bottom, Magenta), both stained for F-Actin using fluorescently conjugated phalloidin (Cyan). The top panel represents a MAX projection, while the bottom image offers a 3D view of the cells. Arrowheads point to cytonemes that are detached from the substrate. Scale bar = 5 μm. (B) Immunofluorescence images of S2 cells stained for endogenous Slik (left, Magenta) and F-Actin (right, Blue), with the right panel showing merged channels. Scale bar = 5 μm. (C) CLSEM images of cells expressing GFP (top) or Slik-GFP (bottom). The left column shows a Z-stack projection of the GFP channel from confocal microscopy. The middle column contains scanning electron microscopy (SEM) images corresponding to the confocal images. The right column shows merged images of confocal microscopy (Magenta) and SEM. An asterisk (*) marks a cell lost during sample preparation for SEM. Scale bar = 10 μm. (D) Graph showing the cytoneme diameter following expression of GFP ($n = 206$) or Slik-GFP ($n = 233$), as measured from SEM micrographs. Each point represents the diameter of an individual cytoneme. (E) Representation of the co-culture experiments protocol. Donor cells were transfected with GFP or Slik-GFP and Hh, and acceptor cells were transfected with Cubitus interruptus (Ci, not expressed by S2 cells and required for Hh signal transduction), and ptc-luciferase. (F) Graph showing the relative luciferase signal emitted by acceptor cells co-cultured with GFP or Slik-GFP donor cells. $n = 16$ from 4 independent experiments, each performed in quadriplicate. (G) Depth color-coded Z-stack images of maximum intensity projections from cells expressing GFP (left) or Slik-GFP (right), excluding the first μm at the base of the cells. Scale bar = 5 μm. (H) Left: Graph depicting the number of cytonemes per cell following expression of GFP ($n = 45$), Slik-GFP ($n = 30$), or after Slik dsRNA treatment ($n = 45$). Each point represents an individual cell. Right: Graph showing cytoneme length (characterized by a length >2 μm and not contacting the substrate) following expression of GFP or Slik-GFP, or after Slik dsRNA treatment. Each point represents the length of an individual cytoneme. $P$-values were calculated using unpaired t-test with Welch's correction (D), Mann–Whitney test (F), unpaired one-way Welch ANOVA with Dunnett T3 (H, left) or Games-Howell (H, right) multiple comparison. Error bars indicate mean ± s.d. Source data are available online for this figure.

## The C-terminal domain of Slik is sufficient to promote cytoneme formation and elongation

Using a kinase dead mutant (Slik[KD]-GFP) (Fig. 4A), we discovered that Slik promotes the formation and elongation of cytonemes in S2 cells independently of its kinase activity (Fig. 4B). Accordingly, we observed that when expressed in wing discs Slik[KD]-GFP also promoted cytoneme formation in DP cells (Fig. 4C).

We previously reported that the association of Slik with the plasma membrane at the apical side of epithelial cells relies on its coiled-coil domain (Panneton et al, 2015). In live discs, Slik[CCD]-GFP localized to dynamic cytonemes projecting from the apical surface of DP cells, most prominently in the marginal zone where neighboring cells make contact (Fig. 4C). These cytonemes crossed the disc lumen and made contact with PerM cells (Fig. 4D). We did not observe the apical blebbing promoted by expression of the full-length kinase. Quantification revealed a large increase in the number of apical cytonemes in Slik[CCD] expressing DP cells compared to controls (Fig. 4D). As with full-length Slik, Slik[CCD]-GFP expression in DP cells was sufficient to drive proliferation of overlying PerM cells (Fig. 4E), consistent with the known dispensability of Slik catalytic activity for this effect (Panneton et al, 2015).

In S2 cells, expression of Slik[CCD] alone was also sufficient to induce both the formation and elongation of cytonemes (Fig. 4B,F) as well as to increase the Hh-response in receiving cells when Slik[CCD] was overexpressed in Hh-donor cells (Fig. EV3). Remarkably, in a large proportion of Slik[CCD]-expressing cells, we observed that Slik[CCD]-GFP formed lateral spikes emanating from the main axis of the cytonemes (Fig. 4F,G). These lateral spikes were formed by an accumulation of Slik[CCD]-GFP, but unlike the body of cytonemes, they were devoid of detectable F-actin (Fig. 4G). Slik[CCD] spikes moved in both directions along cytonemes (Fig. 4H, movie EV2), and when they reached their tip, Slik[CCD] spikes markedly increased the growth speed of cytonemes (Fig. 4H, movie EV2). This suggests that Slik promotes cytoneme elongation through its CCD.

Finally, using CLSEM (Fig. 4I), we observed that Slik[CCD] lateral spikes were formed by tubules of ~950 nm in length and ~165 nm in diameter, a diameter compatible with those of cytonemes (Fig. 4J), suggesting that they may represent the first step of tubulation of the plasma membrane into cytonemes.

## Slik[CCD] sculpts membrane in vitro

Alphafold (Jumper et al, 2021) predicted that Slik[CCD] folds into three bundled alpha-helices (Fig. 5A). Interestingly this is similar to the Bin/amphiphysin/Rvs (BAR) membrane sculpting domains (Simunovic et al, 2019). BAR domains have high affinity for phosphoinositides (Ptdins) and reshape the membranes to which they bind. BAR domains are categorized into classical BARs, N-BARs and F-BARs that generate positive curvatures to form invaginations in cells, and inverse-BARs (I-BARs) that generate negative curvature to initiate cell protrusions. The I-BAR-containing proteins such as Missing-in-metastasis (MIM) or Insulin receptor substrate p53 (IRSp53) also interact with the actin cytoskeleton machinery to promote filopodia growth (Disanza et al, 2013; Kast and Dominguez, 2019; Suetsugu et al, 2006). We thus reasoned that Slik, similar to I-BAR-containing proteins, could sculpt the plasma membrane through its CCD to initiate cytoneme formation and promote their elongation.

Although Slik[CCD] shares similarities with I-BAR domains of MIM or IRSp53, it exhibits unique characteristics. First, Slik[CCD] bundled alpha-helices are predicted to be approximately 1.6 times longer than those of MIM or IRSp53, which would be expected to confer distinct tubulation properties (Fig. 5A). Moreover, while MIM or IRSp53 BAR domains increase the number and length of filopodia when expressed in S2 cells, they fail to form the lateral spikes characteristic of Slik[CCD] (Fig. 5B).

To investigate the direct role of Slik[CCD] as a membrane-sculpting domain, we purified Slik[CCD] and first assessed its ability to dimerize, as is typical for other I-BAR domains. Gel-filtration analysis revealed that Slik[CCD] eluted with a molecular weight consistent with its dimerization (Fig. 5C). We next evaluated its interaction with phosphoinositides using a lipid-binding assay. Unlike classical I-BAR domains that preferably interact with $PI(4,5)P_2$ (Mattila et al, 2007), Slik[CCD] showed strong affinity for Ptdins(4)P and Ptdins (Fig. 5D). Supporting the notion that Slik directly sculpts the membrane to promote cytoneme formation, we found that when added at the exterior of giant unilamellar vesicles (GUV) containing Ptdins(4)P, Slik[CCD] primarily deformed the membranes into tubules extending inward as I-BAR domains do (Fig. 5E,F). More surprisingly, in fewer cases we observed that Slik[CCD] promoted outward tubules, with some GUVs displaying both inward and outward tubules (Fig. 5E,F). In addition, we observed branching on these tubes, a feature never reported before in in vitro experiments

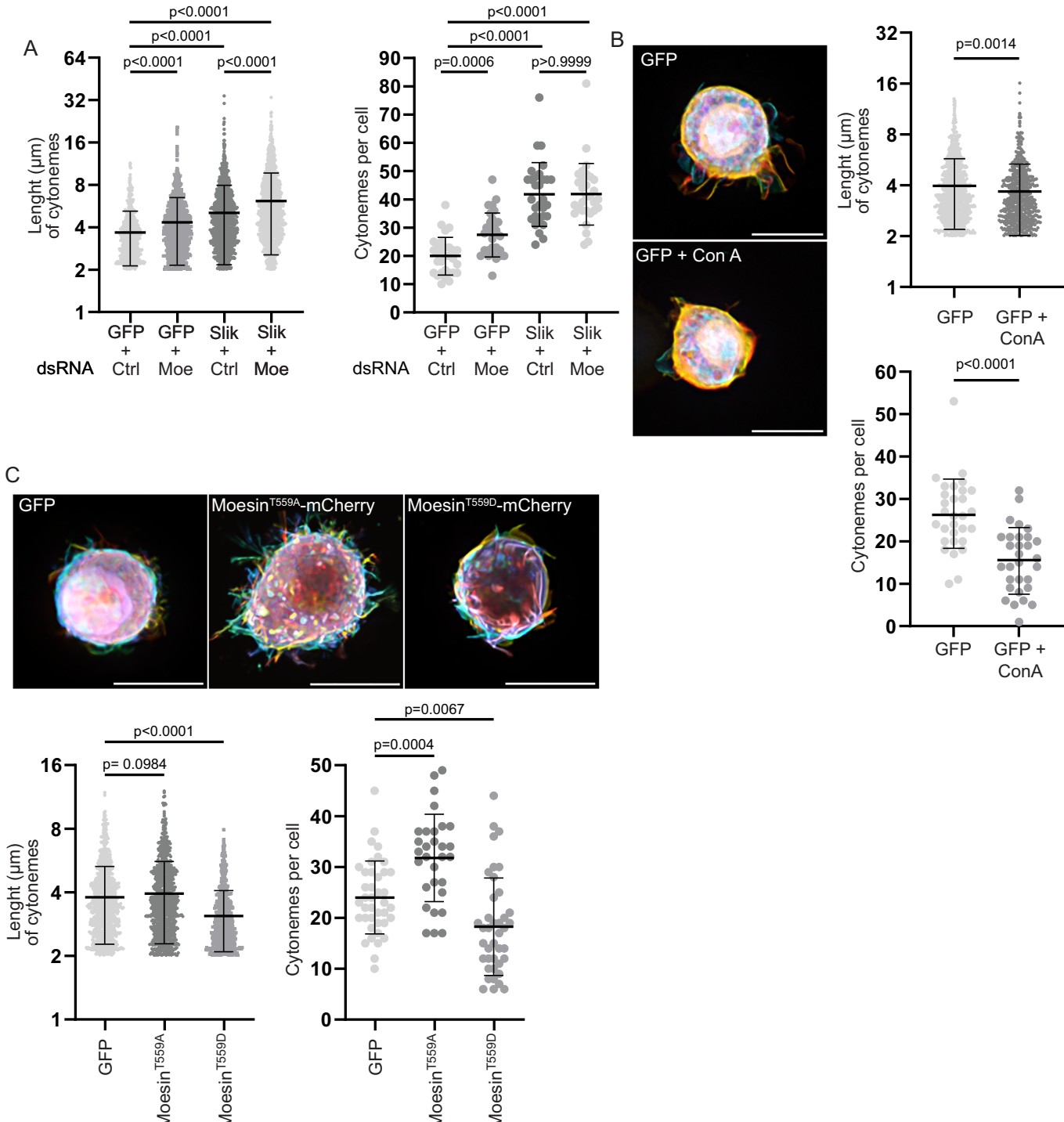

with BAR proteins (Fig. 5E). The ability of Slik[CCD] to generate both negative and positive curvatures on membranes may define Slik as a novel class of membrane sculpting protein.

## STRIPAK controls cytoneme biogenesis by regulating Slik association with the plasma membrane

The phosphorylation of 17 Ser/Thr within its central non-conserved domain (Slik[NCD]) acts as a switch controlling Slik association with the plasma membrane (De Jamblinne et al, 2020). In its non-phosphorylated state, Slik carries a global positive charge, enabling its binding to the negatively charged inner leaflet of the plasma membrane. However, Slik phosphorylation introduces negative charges, leading to its dissociation from the plasma membrane (De Jamblinne et al, 2020). The dSTRIPAK complex plays a crucial role in promoting Slik association with the plasma membrane by dephosphorylating it. We thus wondered whether dSTRIPAK could control cytoneme biogenesis by regulating Slik localization.

**Figure 3. Moesin counteracts cytoneme biogenesis.**

(A) Graphs showing cytoneme length (left) and the number of cytonemes per cell (right) following specified treatments. Each point corresponds to measurements from an individual cytoneme for length, and an individual cell for cytoneme count. $n = 30$ for each condition. (B) Depth color-coded Z-stack images of maximum intensity projections from cells expressing GFP, detailing the effects of treatments on cytoneme dynamics. The left side of the panel shows cells non-treated (top) versus treated with Concanavalin A (Con A) (bottom), excluding the initial micrometer at the cell base. The right side presents quantitative analyses: the top graph displays cytoneme length, and the bottom graph shows the number of cytonemes per cell after the treatments. Each point on the graphs represents measurements from an individual cytoneme (top graph) or an individual cell (bottom graph). $n = 35$ for each condition. Scale bars = 10 μm. (C) The top panel shows depth color-coded Z-stack images of maximum intensity projections from cells expressing GFP (left), Moesin$^{T559D}$-mCherry (middle) of Moesin$^{T559A}$-mCherry (right), excluding the first μm at the cell base. The bottom panel provides a quantitative analysis of cytoneme length and number per cell following the expression of indicated cDNA. Points in the graphs represent measurements from individual cytonemes (left graph) or cells (right graph). GFP $n = 40$, Moesin$^{T559D}$-mCherry $n = 40$, Moesin$^{T559A}$-mCherry $n = 30$. Scale bars = 10 μm. $P$-values were calculated using unpaired t-test with Welch's correction (B), unpaired one-way Welch ANOVA with Dunnett T3 (A, C, right), or Games-Howell (A, C, left) multiple comparison. Error bars indicate mean ± s.d. Source data are available online for this figure.

STRIPAK, an evolutionarily conserved complex, links the phosphatase activity of PP2A to different Ste20-like kinases, including Slik (De Jamblinne et al, 2020). Depletion of dSTRIPAK complex members, such as the Connector of kinase to AP-1 (Cka) regulatory subunit or Striatin interacting protein (Strip), results in Slik dissociation from the plasma membrane (De Jamblinne et al, 2020). Upon dSTRIPAK inactivation, we observed profound effects on cytoneme formation and function. Specifically, Cka dsRNA depletion led to a marked decrease in the number of cytonemes without affecting their length (Fig. 6A,C). Cka depletion also impaired Hh-mediated cell–cell communication in the co-culture assay (Fig. 6B). We further established that dSTRIPAK positively regulates Slik for cytonemes biogenesis, as Cka depletion prevented Slik-GFP from increasing both the number and length of cytonemes (Fig. 6A,C) and accordingly blocked Slik's ability to enhance cell–cell communication in the co-culture assay (Fig. 6B).

To examine whether dSTRIPAK modulates cytoneme biogenesis by regulating Slik association with the plasma membrane, we expressed a non-phosphorylatable mutant of Slik, in which the 17 Ser/Thr of Slik$^{NCD}$ were replaced by Ala (Slik$^{NCD*}$-GFP). In contrast to wild-type Slik, we previously established that Slik$^{NCD*}$ maintained its association to the plasma membrane even when dSTRIPAK is inactivated (De Jamblinne et al, 2020). In addition, we expressed a membrane-localized version of Slik containing a myristoylation (Myr) motif followed by a polybasic (PB) amino acid sequence (MP-Slik-GFP) (Fig. EV4). Expression of either Slik$^{NCD*}$-GFP or MP-Slik-GFP resulted in a comparable increase in both the number and length of cytonemes, similar to the effect observed with Slik-GFP expression (Fig. 6A,C). Notably, dSTRIPAK inactivation did not impede the ability of the non-phosphorylatable form of Slik or the plasma membrane-associated Slik to increase the number and length of cytonemes (Fig. 6A,C), confirming that dSTRIPAK governs cytoneme biogenesis by promoting Slik association with the plasma membrane.

In vivo, we observed a similar requirement of dSTRIPAK for cytoneme formation. Compared to discs expressing Slik-GFP alone, which showed abundant dynamic cytonemes, depletion of *strip* strongly decreased the number of apical cytonemes (Fig. 6D). In time-lapse movies, there was also a decrease in cytoneme dynamics in *strip*-depleted cells (Movie EV3). Although the wing disc exhibits apparent signs of blebbing upon depletion of *strip*, which could suggest apoptosis, we also observe a loss of cytonemes in S2 cells under similar conditions without any evidence of apoptosis such as nuclear fragmentation (Fig. 6C). Consistent with this, our previous

findings demonstrated that dSTRIPAK component depletion in S2 cells neither induced noticeable apoptotic features nor impaired their ability to progress through mitosis (De Jamblinne et al, 2020). This indicates that the blebbing observed in the wing disc and the loss of cytonemes are distinct phenomena that are not mechanistically linked.

Finally, to test if the reduction of cytonemes in dSTRIPAK-depleted cells correlated with an effect on the non-autonomous signaling function of Slik, we quantified Slik-driven non-autonomous proliferation in Strip and Cka depleted wing discs. Compared to control discs expressing Slik with luciferase RNAi, depletion of either Strip or Cka significantly decreased the ability of DP-expressed Slik to drive PerM cell proliferation (Fig. 6E–G). Thus, like its ability to promote cytoneme formation, the ability of Slik to drive proliferation of nearby cells depends on dSTRIPAK.

## Discussion

We discovered that the protein kinase Slik sculpts cytonemes with pro-proliferative functions. Our findings support a model in which dSTRIPAK dephosphorylates Slik to promote its association with the plasma membrane. Through its coiled-coil domain, Slik tubulates the plasma membrane into cytonemes and then promotes their elongation. At the same time, Slik can phosphorylate Moesin, which stiffens the cortex and counteracts cytoneme biogenesis. We propose that this antagonistic interplay, mediated by two domains of the same protein, fine-tunes cytoneme dynamics and their downstream cell–cell communication functions.

Slik membrane-sculpting activity resembles that of I-BAR domain proteins, which promote filopodia formation (Saarikangas et al, 2009; Simunovic et al, 2019). Interestingly, we have recently reported that the I-BAR protein IRSp53 drives formation of tunneling nanotubes, a structure that similarly to cytonemes, is involved in cell–cell communication by physically connecting distant cells (Henderson et al, 2023). However, we identified several differences between Slik and I-BAR domain proteins. To our knowledge, Slik$^{CCD}$ is the first membrane-sculpting domain capable of inducing both negative and positive curvatures on GUVs in vitro. Although many theoretical models have widely investigated membrane-shaping by proteins, none of them has reported such an apparently antagonistic dual property for a single protein, or even more strikingly for the same protein domain. While the negative curvatures are at the origin of cytoneme biogenesis in cells, the potential role(s) of the positive curvatures induced by Slik

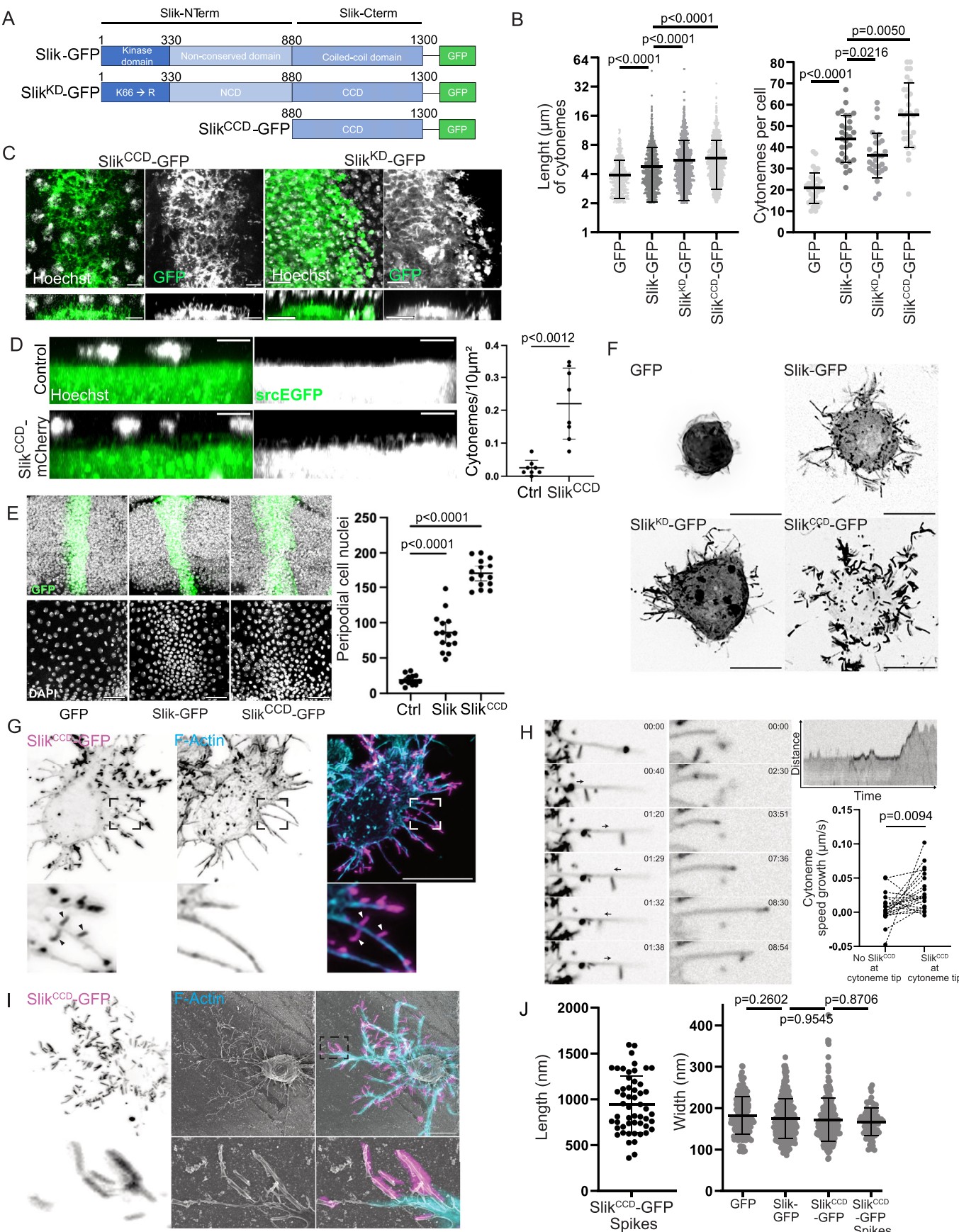

Figure 4. The C-terminal domain of Slik is sufficient to promote cytoneme formation and elongation.

(A). Schematic illustration of Slik protein constructs used in the study. (B) Graphs showing cytoneme length (left) and the number of cytonemes per cell (right) following expression of different Slik constructs. Each point corresponds to measurements from an individual cytoneme (left) or an individual cell (right). $n = 30$ for each condition. (C) Confocal microscopy images showing live wing discs expressing either Slik$^{KD}$-GFP or Slik$^{CCD}$-GFP (Green), with nuclei counterstained using Hoechst (White). The top row depicts MAX projections of XY images capturing both the PerM and the apical DP surface. The bottom row shows MAX projections of XZ images. Scale bars = 10 μm. (D) MAX projection of XZ images from live wing discs expressing srcEGFP alone (top) or together with Slik$^{CCD}$-mCherry (bottom), with Hoechst staining indicating nuclei (White). Graph quantifies the number of cytonemes projecting from the apical DP surface per unit area, based on analysis of 8 discs for each genotype. Scale bars = 5 μm. (E) Confocal images displaying the DP layer (top) and PerM (bottom) of discs expressing GFPnls (left), Slik-GFP (middle), or Slik$^{CCD}$-GFP (right) in the DP layer, with DAPI marking nuclei (White). Scale bars = 20 μm. The graph quantifies the number of PerM nuclei within an 80 μm diameter circle over the wing pouch in wing discs of the indicated genotypes. Analysis was performed on 15 discs of each genotype. (F) MAX projections of Z-stack images from cells expressing the indicated Slik constructs. Scale bars = 10 μm. (G) Immunofluorescence images of S2 cells expressing Slik$^{CCD}$-GFP (Magenta) and stained for F-Actin (Blue), with the right panel showing merged channels. Arrowheads indicate Slik$^{CCD}$-GFP lateral spikes. Scale bars = 10 μm. (H) Time-lapse imaging of a cell expressing Slik$^{CCD}$-GFP, demonstrating lateral movements of Slik$^{CCD}$-spikes along cytonemes (left panels) and their role in promoting cytoneme growth at the tips (right panels). Arrows indicate direction of movement of Slik$^{CCD}$-GFP spikes. The kymograph illustrates the dynamics of the cytoneme shown in the right panels. The graph quantifies the growth speed of same individual cytonemes, when Slik$^{CCD}$-spikes are absent (left) or reached (right) their tip. $n = 20$. (I) CLSEM images of cells expressing Slik$^{CCD}$-GFP. The left row shows a Z-stack projection of the GFP channel from confocal microscopy, the middle row displays scanning electron microscopy (SEM) images corresponding to the confocal images, and the right row presents merged images of confocal microscopy (Slik$^{CCD}$ in Magenta; F-Actin in Blue) with SEM. Scale bars = 5 μm. (J) Quantification of Slik$^{CCD}$-spike length (left, $n = 54$) and width, together with those of the indicated cytonemes (right, $n = 235$ for Slik$^{CCD}$-GFP and $n = 80$ for Slik$^{CCD}$-GFP spikes), as measured from SEM micrographs. Each point represents an individual SlikCCD-extension or cytoneme. Cytoneme width for GFP and Slik-GFP conditions were already shown in Fig. 2D. $P$-values were calculated using unpaired t-test with Welch's correction (D, H), unpaired one-way Welch ANOVA with Dunnett T3 (B, right, E), or Games-Howell (B, left, J) multiple comparison. Error bars indicate mean ± s.d. Source data are available online for this figure.

remains to be investigated. Moreover, we observed that unlike the I-BAR domains in proteins like MIM or IRSp53, the expression of Slik$^{CCD}$ in cells not only promotes formation of cell protrusions but also generates short lateral tubular spikes lacking detectable F-Actin. Remarkably, these spikes are capable of traveling in both directions along cytonemes, indicating that they could be connected to molecular motors. Cells overexpressing full-length Slik also present these tubular spikes, albeit much less frequently, suggesting that they may represent a transient intermediate stage in cytoneme biogenesis, possibly marking the initial step of plasma membrane tubulation into cytonemes. Interestingly, Slik$^{CCD}$ tubular spikes promote rapid elongation of cytonemes when they reach the cytoneme tip, highlighting Slik's dual role in both cytoneme formation and elongation. These two roles bear resemblance to those of I-BAR domain proteins, which scaffold protein complexes that are responsible for actin polymerization, thus facilitating filopodia elongation (Suetsugu et al, 2006; Zhao et al, 2011). However, the specific protein(s) that act in conjunction with Slik to facilitate cytoneme elongation are still to be identified. In addition, we frequently observed vesicles being released from Slik-dependent cytonemes (see Movie EV2). This observation is consistent with findings showing that filopodia induced by I-BAR proteins, such as MIM (Hu et al, 2024; Nishimura et al, 2021), can release extracellular vesicles that stimulate recipient cell migration. While further characterization is required, this suggests that Slik might play a dual role in cell–cell communication, contributing not only to cytoneme biogenesis but also to the production of extracellular vesicles that may carry signaling components.

Several properties of Slik$^{CCD}$ could explain how this domain sculpts membranes differently from I-BAR domains. Its monomeric structure is predicted to be more than 1.5 times longer than other I-BAR domains, a length that could influence its sculpting activity. In addition, this monomeric structure appears flatter than the convex structure of I-BAR proteins like MIM or IrRSp53. We found that Slik$^{CCD}$ binds to Ptdins(4)P, different from I-BAR domains that bind preferably to Ptdins(4,5)P$_2$. Interestingly, both Ptdins(4)P and Ptdins(4,5)P$_2$ are found at the plasma membrane

(Posor et al, 2022) but their respective roles in cytoneme biogenesis have not yet been addressed.

Among the proteins containing a BAR domain, only one family of kinases, FES and FER, combines a BAR domain with a Tyr kinase domain (Carman and Dominguez, 2018). Therefore, Slik represents the first characterized Ser/Thr kinase possessing a BAR-like membrane-sculpting domain. This membrane sculpting function might be conserved, as among the forty-seven human Ste20 kinases, SLK and LOK, the two human orthologues of Slik, as well as Tao1 and Tao3, are predicted to feature a C-terminus with the three bundled alpha-helices characteristic of Slik. Recently, Tao1 was shown to promote the growth of neuron dendritic arbors (Beeman et al, 2023). There was speculation that Tao1 could remodel membranes through its potential BAR domain, although this was not experimentally tested. Slik could thus define a novel family of evolutionary conserved Ser/Thr kinases with membrane sculpting activities that potentially regulate cytoneme formation across species.

We obtained evidence suggesting that the kinase activity of Slik may counteract its membrane sculpting activity within cytonemes. By phosphorylating Moesin, Slik increases cortical rigidity (Kunda et al, 2008). Here we demonstrated that by finely adjusting cortical stiffness through control of the membrane-actin attachment, Moesin can prevent both cytoneme formation and elongation. Supporting this observation, we recently showed that moesin depletion in mammalian osteoclasts decreases membrane-to-cortex attachment and enhances formation of tunneling nanotubes (Dufrancais et al, 2024). The antagonistic roles of Slik$^{CCD}$ and kinase domain could explain why Slik depletion reduces the number of cytonemes but increases their length in S2 cells (Fig. 2H). In these cells without Slik, its CCD cannot tubulate the plasma membrane into nascent cytonemes, leading to fewer cytonemes per cell. However, in the absence of Slik kinase activity, Moesin is not activated, failing to regulate cytoneme length, which may be sustained by other proteins, including additional I-BAR proteins. In an epithelial tissue context, the loss of Slik has a slightly different effect on cytoneme formation that can still be attributed to

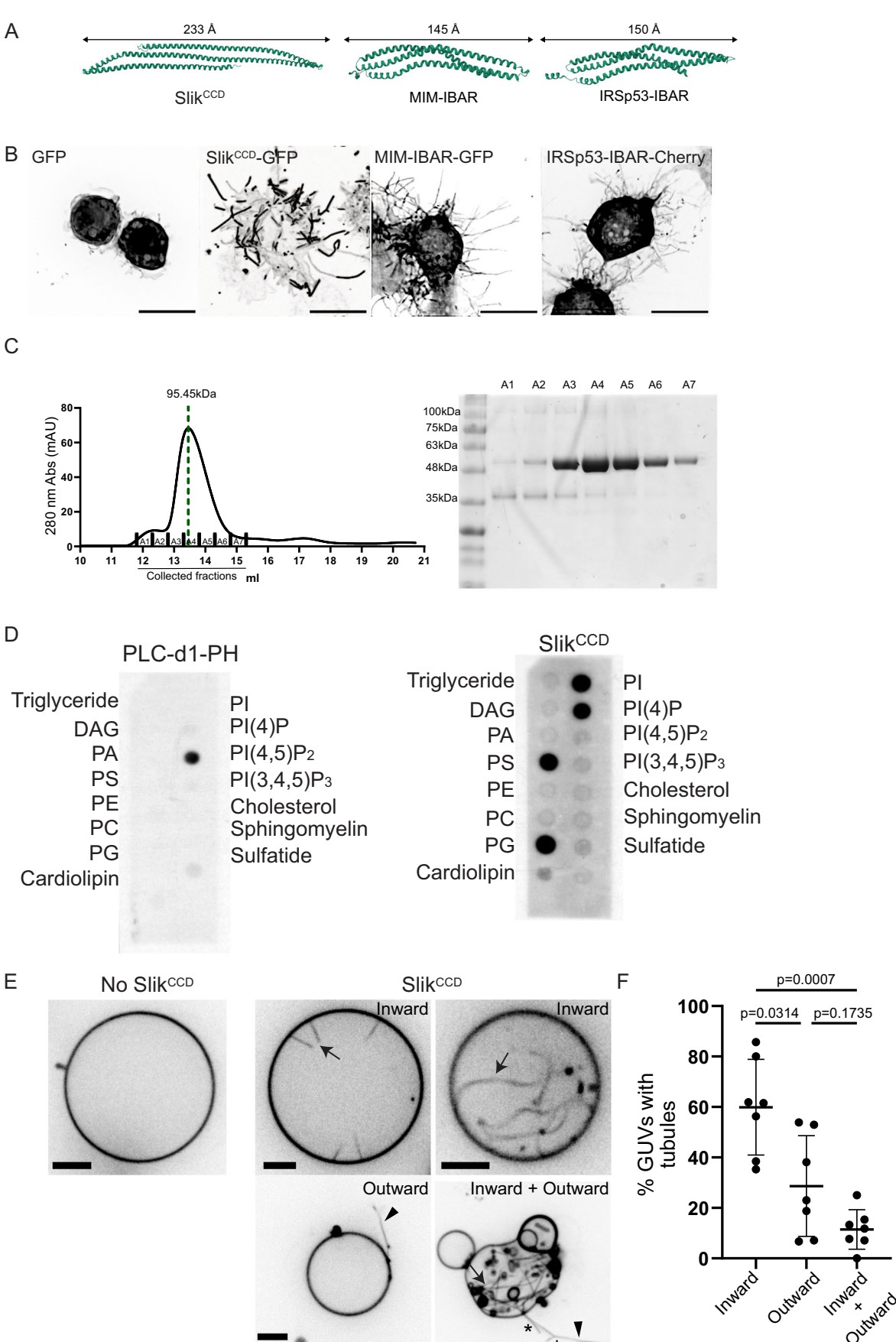

Figure 5.  Slik^CCD sculpts membrane in vitro.

(A) Comparative visualization of protein structures: AlphaFold prediction of Slik^CCD is shown on the left, the I-BAR domain of MIM, as reported by (Millard et al, 2005) is in the middle, and IRSp53, as detailed by (Lee et al, 2007). (B) MAX projections of Z-stack images from cells expressing the indicated constructs. Scale bars = 10 μm. (C) Elution profile of purified Slik^CCD, with an elution peak at 95.45 kDa. Right: SDS-PAGE with coomassie staining of the collected fractions. (D) Evaluation of lipid-protein interactions using lipid strips incubated with either a PI(4,5)P₂-specific binding protein (PLC-delta1-PH, left) or Slik^CCD (right). (E) Visualization of Giant Unilamellar Vesicles (GUVs) containing PtdIns(4)P, both non-incubated (left) and incubated with 0.1 μM or 0.5 μM of Slik^CCD (right, top row and bottom row, respectively). Arrows indicate the formation of inward tubules, arrowheads indicate outward tubules and stars indicates branched tubules generated by the interaction with Slik^CCD, underscoring its potential role in membrane deformation. Scale bars = 5 μm. The images capture fluorescent signals from GUV membranes tagged with BODIPY-TR-C5-ceramide, presented in inverted grayscale to enhance tubulation visualization. (F) Percentage of PtdIns(4)P GUVs displaying inward, outward, or both inward + outward tubules when incubated with Slik^CCD at concentrations ranging from 0.1 to 1 μM. N = 7 experiments, and n = 21, 13, 26, 17, 14, 16, 30 total amount of GUVs in the 7 experiments. Error bars indicate mean ± s.d. Source data are available online for this figure.

the antagonistic function of its catalytic activity. We propose that loss of Moesin activity at the apical membrane in the absence of Slik, and the consequent reduction of membrane stiffness, facilitates the formation of apical protrusions through alternative mechanisms. However, without Slik, these protrusions lack signaling function. While we have not yet identified the mechanism regulating Slik activity towards its membrane-sculpting or kinase functions, it is tempting to speculate that a molecular switch coordinating these two functions could govern cytoneme formation and elongation.

In addition, we have discovered that the dSTRIPAK complex also plays an essential role in cytoneme formation, at least in part, by regulating Slik association with the plasma membrane. We previously demonstrated that the phosphatase activity of dSTRI-PAK reduces Slik phosphorylation, promoting its cortical association and proper activation of Moesin, a mechanism that controls mitotic morphogenesis and epithelial integrity (De Jamblinne et al, 2020). Here, we observed that Slik depletion leads to a reduction in the number of cytonemes expressed by Drosophila cells in culture, confirming the important role of Slik in controlling the biogenesis of these structures. However, we also noticed that the inactivation of dSTRIPAK reduces the number of cytonemes per cell more drastically than the depletion of Slik alone, suggesting that dSTRIPAK may regulate other proteins that play important roles in cytoneme biogenesis.

Our results suggest that Slik promotes the delivery of growth signals from DP to PerM cells by increasing the number and length of apical cytonemes. Such transluminal signaling has been previously observed in wing discs. For instance, Dpp and Notch ligands as well as signals downstream from EGF in PerM cells were shown to stimulate responses in the DP (Gibson et al, 2002; Gibson and Schubiger, 2000; Pallavi and Shashidhara, 2003; Paul et al, 2013). Conversely, Dpp expressed in DP cells seemed to activate signaling in the PerM (Gibson et al, 2002). Furthermore, apical actin-based DP cell protrusions that appear to contact PerM cells, similar to those that we observed, have previously been described in wing discs. However, their link to transluminal signaling is unclear (Demontis and Dahmann, 2007). Our GFP complementation approach confirmed that transluminal membranes connect DP to PerM cells in control discs. Furthermore, we found that Slik overexpression in the DP increases the number and length of apical cytonemes and connection to the PerM epithelium. This correlated with an increase in PerM cell proliferation. These observations suggest that cytonemes mediate direct contact between the two epithelial layers to permit growth signal exchange

in wing discs. Interestingly, PerM size scales with increased or decreased DP size, suggesting the existence of a feedback mechanism that coordinates the two epithelia (Pallavi and Shashidhara, 2003). Signal exchange mediated by apical cytonemes formed under the control of the dSTRIPAK-Slik axis could therefore provide a mechanism that allows DP and PerM epithelia to match their size with each other.

Our initial focus on the apical plasma membrane came from the observation that Slik is enriched apically in wing discs (Hipfner et al, 2004), where the most prominent phenotype, DP-PerM signaling, was apparent. However, cytonemes have previously been observed emanating from the basolateral region of the wing disc (Wood et al, 2021), where they mediate signaling pathways such as Hedgehog (Hh) and Fibroblast Growth Factor (FGF; Branchless) (Chen et al, 2017; Du et al, 2022). Analysis of cryosectioned wing discs revealed that Slik and its phosphorylated substrate Moesin localize to the basolateral membranes of DP cells, with notable enrichment at the basal surface (Fig. EV1). Interestingly, Slik-GFP and Slik^CCD-GFP were also detected in filopodia-like projections at the basal surface of the wing disc (Fig. EV5A,B). In addition, when expressed in the air-sac primordium, a tissue adjacent to the wing disc where cytoneme function is well-characterized (Roy et al, 2011), Slik^CCD-mCherry associated with basolateral cytonemes (Fig. EV5C). Interestingly, beyond its role in cytoneme initiation and extension, Slik may also contribute to their stabilization, as suggested by the increased persistence of these structures when Slik is overexpressed. Together, these findings suggest that Slik plays a broader role in cytoneme function in Drosophila, which remains to be further investigated.

The biogenesis of membrane protrusions and tubes, organized by the actin cytoskeleton, is essential to many specialized cellular functions. A common theme emerging from the analysis of Slik and its mammalian orthologs SLK and LOK is their involvement in the formation of such membrane tubules. In Drosophila, Slik is required for the formation of the photosensitive rhabdomeres of photoreceptors. These tightly packed arrays of hundreds of actin-based tubular microvilli are highly disorganized in the absence of Slik (Hipfner and Cohen, 2003; Ogi et al, 2019). Slik mutants also show markedly reduced complexity of subcellular tube branching in terminal cells of the tracheal system, whose formation is orchestrated by the actin cytoskeleton (JayaNandanan et al, 2014; Ukken et al, 2014). In mammals, loss of SLK in cortical neurons leads to reduced branching of distal dendritic arbors, whereas its overexpression increases this branching (Schoch et al, 2021). Knockout of SLK in kidney podocytes disrupts the finely

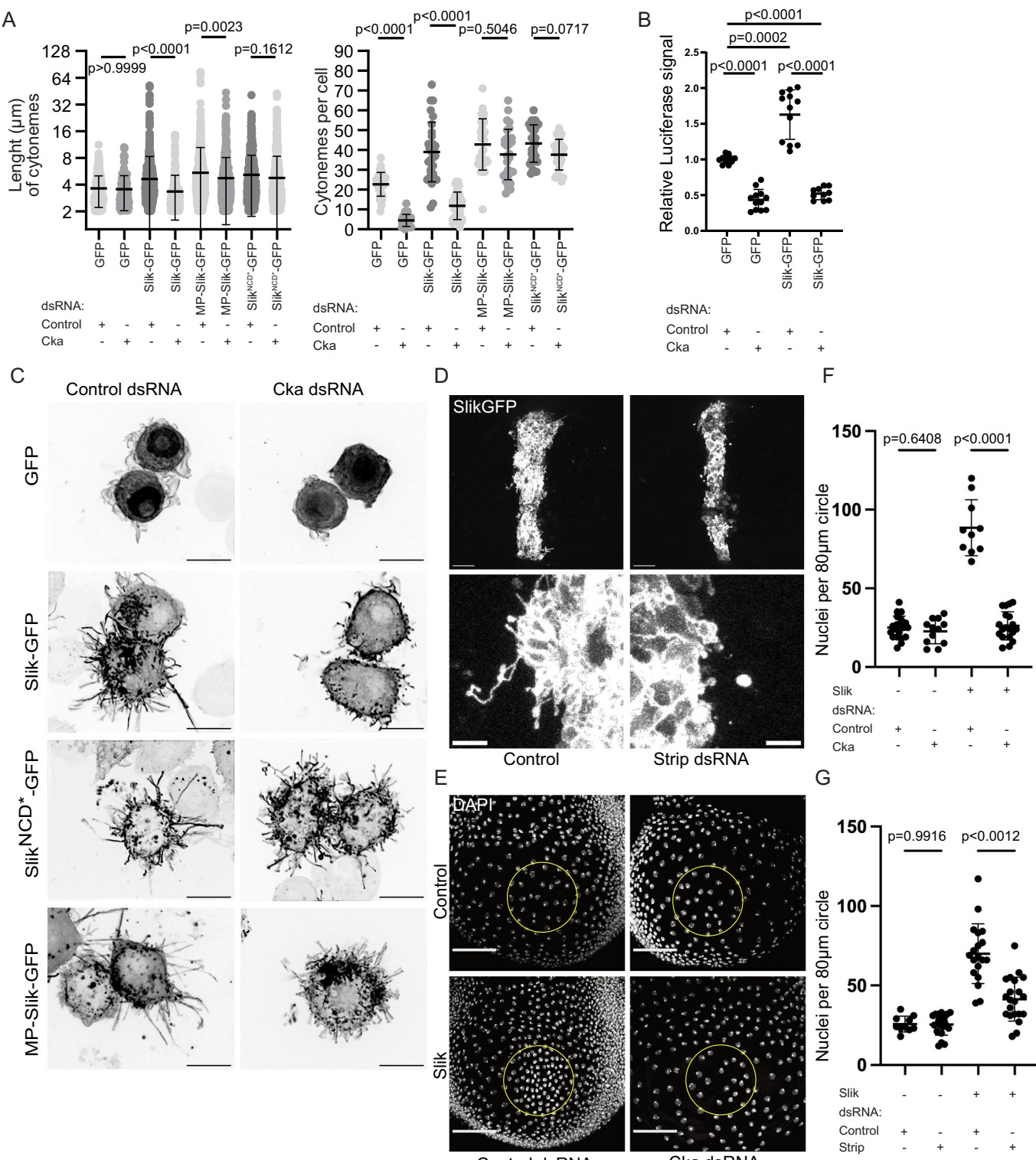

interdigitating foot processes in the glomerulus (Cybulsky et al, 2018). The generation of both dendritic complexity and foot process architecture is dependent upon actin-based shaping of membranous tubes (Konietzny et al, 2017; Welsh and Saleem, 2011). While misregulation of ERM protein activation likely makes an important contribution to the defects observed in Slik/SLK mutants (JayaNandanan et al, 2014; Karagiosis and Ready, 2004), our findings point to and underscore a more general role of Slik in sculpting membranes to control the biogenesis of a diversity of specialized membrane protrusions.

◄ **Figure 6. STRIPAK controls cytoneme biogenesis by regulating Slik association with the plasma membrane.**

(A) Graphs show cytoneme length (left) and the number of cytonemes per cell (right) under the indicated treatments. Each data point represents the measurement from an individual cytoneme (left) or cell (right). $n = 30$ for each condition. (B) Graph showing the relative luciferase signal emitted by acceptor cells co-cultured with GFP or Slik-GFP donor cells and treated with Cka or control dsRNA. $n = 16$ from 4 independent experiments, each performed in quadruplicate. (C) MAX projections of Z-stack images from cells expressing the indicated Slik constructs upon dsRNA treatment against Gal4 (left column) or Cka (right column). Scale bars = 10 μm. (D) MAX projections of Z-stack images through the apical side of DP cells expressing Slik-GFP alone (left) or together with RNAi targeting Strip (right). Scale bars = 20 μm (top images) and 5 μm (bottom images). (E) Confocal images showing the PerM of wing discs expressing RNAi against Luciferase (left) or Cka (right) in the DP layer, alone (top row) or in combination with Slik expression (bottom row). Nuclei are stained with DAPI (White). Scale bars = 50 μm. (F) Graph shows quantification of PerM nuclei within an 80 μm diameter circle in wing discs expressing Luciferase or Cka dsRNA as in (D), across various genotypes. Analysis was performed on (from left to right) 20, 12, 10, and 19 discs. (G) Graph shows quantification of PerM nuclei within an 80 μm diameter circle in wing discs expressing Luciferase or Strip dsRNA, across different genetic backgrounds. Analysis was performed on (from left to right) 10, 20, 19, and 21 discs. *P*-values were calculated using unpaired one-way Welch ANOVA with Dunnett T3 (A, right, F, G) or Games-Howell (A, left, B) multiple comparison. Error bars indicate mean ± s.d. Source data are available online for this figure.

# Methods

### Reagents and tools table

| Reagent/Resource | Reference or Source | Identifier or Catalog Number |
|---|---|---|
| **Experimental models** | | |
| Drosophila S2 cells | Schneider, 1972 | |
| Ptc-GAL4 | BDSC | 2017 |
| Nub-GAL4 | BDSC | 86108 |
| Tub-GAL80ts | BDSC | 7108 |
| Ap-GAL4 | BDSC | 3041 |
| UAS-GFPnls | BDSC | 4775 |
| UAS-LifeAct-RFP | BDSC | 58362 |
| UAS-srcEGFP.M7E | BDSC | 5432 |
| UAS-CD4-spGFP1-10 | BDSC | 93017 |
| LexAop-CD4-spGFP11 | BDSC | 93018 |
| UAS-luc RNAi | BDSC | 35788 |
| EP-Slik (Slik20348) | Panneton et al, 2015 | |
| Crb-GFP | VDRC | 318462 |
| Cka RNAi | VDRC | 106971 |
| Strip RNAi | VDRC | 106184 |
| Slik RNAi | VDRC | 43783 |
| UAS-Slik-GFP | Panneton et al, 2015 | |
| UAS-SlikKD-GF | Panneton et al, 2015 | |
| UAS-SlikCCD-GFP | Panneton et al, 2015 | |
| UAS-SlikCCD-mCherry | Panneton et al, 2015 | |
| **Recombinant DNA** | | |
| pGEX-6P-1 | Amersham | 27-4597-01 |
| **Antibodies** | | |
| Guinea pig anti-Slik | Hipfner and Cohen, 2003 | |
| Rabbit Anti-GFP | OriGene Technologies | TP401 |

| Reagent/Resource | Reference or Source | Identifier or Catalog Number |
|---|---|---|
| Rabbit Anti-Phospho-Ezrin/Radixin/Moesin | Cell Signaling Technology | 3276 |
| Anti-guinea pig-A488 | Invitrogen | A11073 |
| Mouse GST Tag Antibody | Invitrogen | MA4-004 |
| Rabbit Slik CCD antibody | Jamblinne et al, 2020 | |
| Anti-mouse-Hrp antibody | Santa Cruz | Sc-516102 |
| Anti-Rabbit Hrp antibody | Jackson ImmunoResearch | 111-035-144 |
| Armadillo antibody | DSHB | AB_528089 |
| **Oligonucleotides and other sequence-based reagents** | | |
| DsRNA for S2 cells treatment | De Jamblinne et al, 2020 | |
| **Chemicals, Enzymes and other reagents** | | |
| Schneider's Drosophila medium | GIBCO | 21720001 |
| FBS | Invitrogen | 12484028 |
| penicillin-streptomycin | GIBCO | 15140122 |
| FuGENE HD Transfection Reagent | Promega | E2311 |
| Opti-MEM | GIBCO | 31985070 |
| Egg phosphatidylcholine | Sigma | 840051 |
| brain phosphatidylinositol-4-phosphate | Sigma | 840045 |
| L-α-lysophosphatidylinositol | Sigma | 850091 |
| 1,2-dioleoyl-sn-glycero-3-phospho-L-serine | Sigma | 840035 |
| β-casein | Sigma | C6905 |
| BODIPY-TR-C5-ceramide | Invitrogen | D7540 |
| Culture-Inserts 2 Well | Ibidi | 80209 |
| PFA | Thermo Fisher Scientific | 043368-9M |
| Glutaraldehyde | Thermo Scientific | AAA1050036 |
| BSA | NEB | B9000S |
| Texas-red Phalloidin | Invitrogen | T7471 |
| Mowiol | Sigma-Aldrich | 81381 |
| OsO4 | EMS | 19150 |

| Reagent/Resource | Reference or Source | Identifier or Catalog Number |
|---|---|---|
| Luciferase assay reagent | Promega | E151A, E152A |
| Concanavalin A | Sigma-Aldrich | C7275 |
| Lipid strips | Echelon bioscience | P-6002 |
| Shield and Sang M3 insect Medium | Sigma | S8398 |
| Hoechst 33342 | ThermoScientific | H3570 |
| DAPI | Invitrogen | 62248 |
| Cryomold | Fisherbrand | 22-363-553 |
| OCT | Fisher Healthcare | 4585 |
| **Software** | | |
| FIJI | Schindelin et al (2012) | |
| Icy | de Chaumont et al (2012) | |
| **Other** | | |
| Critical point dryer CPD300 | Leica | |
| gridded 35 mm dish | IBIDI | 81148 |
| Zeiss observer z1 | Zeiss | |
| EM ACE 600 | Leica | |
| Regulus 8220 | Hitachi | |
| Infinite M200 Pro | Tecan | |
| Eclipse Ti-E | Nikon | |
| LSM 700 | Zeiss | |
| Cryostar NX70 | Thermo Fisher Scientific | |

## S2 cell culture and cDNA/dsRNA transfection or drug treatments

Drosophila S2 cells were grown at 25 °C in Schneider's Drosophila medium (21720001; GIBCO) complemented with 10% FBS (12484028; Invitrogen) and 1% penicillin-streptomycin antibiotics (15140122; GIBCO). All plasmid cDNA transfections were performed using the FuGENE HD Transfection Reagent (E2311; Promega) in Opti-MEM (31985070; GIBCO) media and 1 μg cDNA was used for $1.5 \times 10^6$ cell transfection. Cells were transfected for 48 h before imaging. Knockdown experiments were performed by plating $1.5 \times 10^6$ cells in a 6-well plate and incubating with 15 μg dsRNA for 5 days. dsRNA production protocol and sequences are described in De Jamblinne et al (2020). Egg phosphatidylcholine (EPC, 840051), brain phosphatidylinositol-4-phosphate (PI4P, 840045), L-α-lysophosphatidylinositol (LPI, 850091), 1,2-dioleoyl-sn-glycero-3-phospho-L-serine (DOPS, 840035), and β-casein from bovine milk (>98% pure, C6905) were purchased from Sigma. BODIPY-TR-C5-ceramide, (TR ceramide, D7540) was purchased from Invitrogen. Culture-Inserts 2 Well for self-insertion were purchased from ibidi (Silicon open chambers, 80209).

## Immunofluorescence of S2 cells

$5 \times 10^5$ cells were seeded on glass coverslips in 0.5 ml of medium. After 1 h cells were fixed in MEM-fixation (4% PFA, 0.5% glutaraldehyde, 0.1 M Sorenson's phosphate buffer at pH 7.4), at room temperature for 7 min as in (Bodeen et al, 2017). Cells were then washed with TBS and 1 mg/ml NaBH4 for 10 min to reduce auto fluorescence, followed by two 10 min TBS washes at room temperature. Cells were then permeabilized and blocked with TBS-Triton 0.1% - BSA 1% for 1 h at room temperature. Primary antibody was applied in fresh blocking solution. Guinea pig anti-Slik (Hipfner and Cohen, 2003) was used at 1/2000 and incubated overnight at room temperature. Cells were washed three times with TBS. The secondary antibody was Alexa Fluor 488-conjugated anti-guinea pig antibody (A11073; Invitrogen - 1/100). We incubated cells with Texas Red-X Phalloidin (T7471; Invitrogen - 1/200) for 30 min at room temperature to stain F-actin. Stained cells were washed three times with TBS, and coverslips were dried and mounted in Mowiol (81381; Sigma-Aldrich). Immunofluorescence images were acquired with a Spinning disk (Zeiss observer Z1) equipped with a Plan Apochromat 63x objective/NA 1.4, a Zeiss Axiocam 506 Mono camera and a CSU-X1 confocal scanning unit, using Zen software (ZEISS Microscopy). Representative images were rescaled for publication using Photoshop (Adobe).

## Correlative light electron microscopy

Cells were grown on IBIDI gridded 35 mm dish (Cat.No:81148), fixed as for immunofluorescence, stained by Phalloidin-Texas Red 30 min at room temperature, followed by three wash in phosphate buffer. Acquisition was performed on a Leica SP8 confocal microscope with a Plan Apochromat 63x/NA 1.4 objective, with additional acquisition of bright field images for image registration in electron microscopy. After confocal acquisition cells were fixed in glutaraldehyde 2.5% (AAA1050036; Thermo Scientific) overnight at 4 °C, washed three times with Phosphate Buffer and post-fixed in $OsO_4$ 1% (19150; EMS) for 1 h on ice, followed by three washes in $dH_2O$. Dehydration of the sample was performed by successive bath in increasing concentration of ethanol (30%, 50%, 70%, 80%, 90%, 95% and two times 100%). Coverslips were then detached from the dish with a glass scribe equipped with a tungsten carbide tip. Evaporation was realized with a Critical Point Dryer (Leica CPD300), followed by 5 nm carbon coating (Leica EM ACE600). Scanning electron microscopy was performed with a Hitachi Regulus 8220 with a cold field emission source, with the SE(L) mode at 1kv 10 μA, or LA BSE at 0.9 kv 10 μA to reduce artefacts due to charge accumulation. Correlation between confocal and electron microscopy images was performed using the ec-CLEM plugin with the Icy software (Institut Pasteur).

## Live cell imaging and time-lapse microscopy

For time-lapse microscopy and live cell imaging, $5 \times 10^4$ cells were plated in a 96-well plate (655892; Greiner Bio-One), 48 h before imaging. Acquisition was performed with a Spinning disk (Zeiss observer Z1) equipped with a Plan Apochromat 63x objective/NA 1.4, a Zeiss Axiocam 506 Mono camera and a CSU-X1 confocal scanning unit, using Zen software (ZEISS Microscopy) at room temperature, while cells were in Schneider's Drosophila medium. For time-lapse imaging, cells were imaged every 3 s for 10 min with a low laser intensity. Images were analyzed and quantified with ImageJ software (NIH). Representative images were rescaled for publication using Photoshop (Adobe).

## Co-culture experiments

On day 1, $7.5 \times 10^5$ donor and acceptor cells were plated in a 24-well plate (82050-892; Greiner Bio-One) and transfected with plasmid encoding for Hedgehog and GFP/Slik-GFP/Slik$^{CCD}$-GFP (donor cells) or Ptc-luciferase and cubitus interruptus (acceptor cells). On day 2, the cells were re-suspended in fresh media, combined at a ratio 1:3 of acceptor to donor cells, and plated in a 96-well plate (82050-771; Greiner Bio-One) in quadriplicates. On day 4, cells were washed with PBS and lysed with 40 µL of lysis buffer (E153A; Promega) for 15 min. Then, 12 µL of the lysate were transferred to a white-walled 96-well plate (165306: Thermo Scientific) followed by the addition of 30 µL of Luciferase assay reagent (E151A, E152A; Promega) and the plate was read using an Infinite M200 Pro (Tecan).

## Concanavalin A coating experiments

Cells were re-suspended for 30 min with or without Concanavalin A (C7275; Sigma-Aldrich) at 15 µg/mL and $0.1 \times 10^6$ cells in 200 µL were plated in 96-well plate (655892; Greiner Bio-One) 1 h before imaging.

## Image analysis and cytoneme quantification

Images were analyzed using FIJI (NIH) using a custom macro adapted from (Barbosa and Kornberg, 2022). Briefly, we removed the first 1 µm at the bottom of the cell and performed maximum intensity projection on deconvoluted images (DeconvolutionLab2, with a measured PSF and RL algorithm), with a color coding corresponding to the Z position of every pixel from the source image. This allowed us to visually distinguish overlapping protrusions within each cell and perform quantification by highlighting every cytonemes by hand. For 3D representation, we used the ImageJ plugin ClearVolume.

## Slik$^{CCD}$ purification

Slik$^{CCD}$ was cloned in a pGEX-6P-1 plasmid (27-4597-01; Amersham) using EcoRI and NotI restriction sites. BL21(DE3) (EC0114; Thermo Scientific) were transformed and cultivated according to product guidelines. After Induction with 0.5 M IPTG (AM9462; Invitrogen) overnight at 16 °C and bacteria lysis, the soluble fraction was incubated with gluthatione agarose beads (L00206: GenScript) for 2H. After washes, cleavage of the GST tag was performed using PreScission Protease (27-0843-01; Cytiva) overnight at 4 °C.

## Gel filtration

After purification Slik$^{CCD}$ was subjected to size exclusion chromatography using a Superdex 75 10/300 column (GE Healthcare) in Tris-based buffer (20 mM Tris-HCl pH 7.5, 150 mM NaCl, 2 mM DTT).

## Lipid strip experiments

Slik$^{CCD}$ was incubated with lipid strips (P-6002; Echelon Biosciences) following manufacturer recommendations. PI(4,5)P2 Grip (G-4501; Echelon Bioscience) was used at 0.5 µg/mL as a control, and Slik$^{CCD}$

domain was used at 1 µg/mL. To detect PI(4,5)P2 a GST Tag Antibody (MA4-004; Invitrogen) was used, and a Slik antibody (De Jamblinne et al, 2020) was used to detect Slik$^{CCD}$. Secondary antibody coupled to HRP directed against Mouse (sc-516102; Santa Cruz) and Rabbit (111-035-144; Jackson ImmunoResearch) were then used.

## Giant unilamellar vesicles (GUV) preparation

GUVs were generated using a lipid mixture consisting of EPC/10 mol% DOPS/10 mol% PI4P/0.5 mol% TR ceramide dissolved at 1 mg/mL in chloroform. The internal buffer used to prepare GUVs was 50 mM NaCl, 20 mM sucrose and 20 mM Tris pH 7.5. GUVs were prepared using the polyvinyl alcohol (PVA) gel-assisted vesicle formation method as described previously (Weinberger et al, 2013). A PVA gel solution (5%, w/w, dissolved in 280 mM sucrose and 20 mM Tris, pH 7.5) was warmed up to 50 °C. The warm PVA solution was spread on coverslips (20 mm × 20 mm), which were cleaned with ethanol and ddH$_2$O before use. The PVA-coated coverslips were dried at 50 °C for 30 min. Approximately 5 µL of the lipid mixture was spread on the PVA-coated coverslips. The lipid-coated coverslips were vacuumed for 30 min at room temperature to remove chloroform. The coverslips were then placed in petri dishes and 500 µL of the inner buffer was pipetted onto each coverslip. The coverslips were kept at room temperature for at least 45 min to allow the GUVs to grow. Once this was done, we gently "ticked" the bottom of the petri dish to detach GUVs from the PVA gel. The GUVs were collected using a 1 mL pipette tip with the tip cut off to avoid breaking the GUVs.

## Sample preparation and observation of GUVs incubated with Slik$^{CCD}$

Experimental chambers were assembled by placing the silicon open chamber on a coverslip, which was cleaned with ethanol and ddH$_2$O before use. The chamber was passivated with a β-casein solution at a concentration of 5 mg/mL for at least 5 min at room temperature. To mix GUVs with Slik CTD, we first added 5 µl of CTD (5 mM in stock, diluted with the outer buffer if needed) in 25 µL of the outer buffer (60 mM NaCl and 20 mM Tris, pH 7.5), followed by adding 20 µL of the GUV solution. GUVs were incubated with Slik CTD in the experimental chamber for at least 30 min at room temperature before observation. Samples were observed using a spinning disk confocal microscope, Nikon eclipse Ti-E equipped with a Yokogawa CSU-X1 confocal head, 100X CFI Plan Apo VC objective (Nikon) and a CMOS camera Prime 95B (Photometrics).

## Plasmid construction for transgenic fly generation

Slik, Slik$^{KD}$ (= Slik$^{D176N}$) and Slik$^{CCD}$ coding sequences were cloned with an in-frame C-terminal GFP tag into pUAST for making transgenic flies. For pUAST/Slik-GFP and pUAST/Slik$^{KD}$-GFP, an MluI restriction site was introduced immediately upstream of the stop codon and XhoI site in pBluescript plasmids bearing full-length Slik and Slik$^{KD}$ coding sequences (PA isoform) as EcoRI-XhoI fragments (Hipfner and Cohen, 2003) by QuickChange mutagenesis (Agilent Technologies). For pUAST/Slik$^{CCD}$-GFP, the Slik$^{CCD}$ coding sequence (encoding amino acids 889–1300 from the C-terminus of Slik-PA) was PCR amplified from pBS/Slik,

introducing an upstream EcoRI site followed by an initiator ATG codon, and a MluI site immediately upstream of the stop codon, and the resulting product cloned into pGEM-T. These EcoRI-MluI insert fragments were then cloned together with a MluI-XhoI fragment encoding the C-terminal GFP tag (generated by PCR) into pUAST digested with EcoRI and XhoI. pUAST/Slik^CCD-mCherry was prepared as above, using a MluI-XhoI fragment encoding a C-terminal mCherry tag (generated by PCR).

To create a PerM-specific LexA driver strain, we first searched the Janelia FlyLight Project database for genomic fragments with PerM expression in wing discs (Jory et al, 2012). Among the candidates, we identified one fragment (GR31F05) mapping upstream of the *stg* gene that showed highly-specific wing disc PerM expression in late third instar. We PCR-amplified the 31F05 genomic fragment using the oligos 5'-ctgcccaaatccagtgcaatgtaaa-3' and 5'-ctgaattttgggagttggtcccgtc-3'. The resulting 3060 nucleotide fragment was cloned into the pCR8/GW/TOPO backbone (Thermo Fisher Scientific). This insert was then recombined by Gateway cloning into pBPnlsLexA::p65Uw (Pfeiffer et al, 2010) to create pBP-PerM-nlsLexA::p65Uw.

## Drosophila stocks and culture

*nubGAL4, ptc-GAL4* (*ptc^559.1*),*tub-GAL80^ts*, *apGAL4*, UAS-GFPnls, UAS-LifeAct-RFP (#58362), UAS-srcEGFP.M7E (#5432), UAS-CD4-spGFP1-10 (#93017), LexAop-CD4-GFP11 (#93018), and UAS-luc RNAi strains (#35788) strains were obtained from the Bloomington *Drosophila* Stock Center. EP-Slik (*slik^20348*) is an EPg element insertion upstream of the *slik* gene that drives its GAL4-dependent expression. Crb-GFP (#318462) and transgenic dsRNA strains targeting *cka* (#106971), *strip* (#106184), and *slik* (#43783) were obtained from the Vienna Drosophila Resource Center.

Transgenic strains were generated by Genome ProLab (Sherbrooke, QC, Canada). *UAS-Slik-GFP, UAS-Slik^KD-GFP, UAS-Slik^CCD-GFP*, and *UAS-Slik^CCD-mCherry* strains were generated by standard P-element-mediated transformation into *w^1118*. PerM-LexA was generated by recombination into the attP2 landing site on chromosome 3 (at 68A4).

Crosses were performed at 25 °C unless stated otherwise. The genotypes were as follows:

w;*ptcGAL4/+;UAS-LifeActRFP/+* (Fig. 1A), w;*ptcGAL4/EP-Slik;UAS-LifeActRFP/+* (Fig. 1A), w;*apGAL4/Crb-GFP* (Fig. 1B), w;*apGAL4,EP-Slik/Crb-GFP* (Fig. 1B), w;*ptcGAL4,UAS-srcEGFP/+* (Figs. 1C and 4D), w;*ptcGAL4,UAS-srcEGFP/EP-Slik* (Fig. 1C), w;*ptcGAL4/+;UAS-Slik-GFP/tub-GAL80^ts* at 18 °C then 48 h at 27 °C (Figs. 1D,E, 4E and 6D), w; *nubGAL4/UAS-CD4-spGFP1-10;LexAop-CD4-spGFP11/+* (Fig. 1G, left), w;*nubGAL4/UAS-CD4-spGFP1-10;LexAop-CD4-spGFP11/PerM-LexA* (Fig. 1G, middle), w;*nubGAL4,EP-Slik/UAS-CD4-spGFP1-10;LexAop-CD4-GFP11/PerM-LexA* (Fig. 1G, right), w;*ptcGAL4/+;UAS-Slik^CCD-GFP/+* (Fig. 4C,E), w;*ptcGAL4/+;UAS-Slik^KD-GFP/+* (Fig. 4C), w;*ptcGAL4,UAS-srcEGFP/UAS-Slik^CCD-mCherry* (Fig. 4D), w;*ptcGAL4/UAS-GFPnls* (Fig. 4E), w;*ptcGAL4/+;UAS-Slik-GFP,UAS-strip.dsRNA/tub-GAL80^ts* at 18 °C then 48 h at 27 °C (Fig. 6D), w;*ptcGAL4/+;UAS-luc.dsRNA/+* (Fig. 6E), w;*ptcGAL4/+;UAS-cka.dsRNA/+* (Fig. 6E), w;*ptcGAL4,EP-Slik/+;UAS-luc.dsRNA/+* (Fig. 6E), w;*ptcGAL4,EP-Slik/+;UAS-cka.dsRNA/+* (Fig. 6E), w;*ptcGAL4/+;UAS-luc.dsRNA/+* (graph Fig. 6G), w;*ptcGAL4/+;UAS-strip.dsRNA/+* (graph Fig. 6G), w;*ptcGAL4,EP-Slik/+;UAS-luc.dsRNA/+* (graph Fig. 6G), w;*ptcGAL4,EP-Slik/+;UAS-strip.dsRNA/+* (graph Fig. 6G).

For analysis of fixed samples, wandering third instar larvae were dissected and anterior halves with attached wing discs were collected in PBS on ice. Discs were fixed in PBS containing 0.2% Tween-20 (PBT) and 4% formaldehyde for 20 min. After several washes with PBT, samples were blocked in PBT with 0.1% BSA, followed by incubation overnight at 4 °C with primary antibodies against GFP (1:100, TP401; OriGene Technologies) (Fig. 1B). Samples were incubated with Fluor-conjugated secondary antibodies (Invitrogen and Jackson ImmunoResearch Laboratories) and DAPI (0.5 µg/ml, Invitrogen). Discs were mounted on slides in mounting medium (10% PBS, 90% glycerol, 0.2% n-propyl gallate), coverslipped, and imaged at RT using a Zeiss LSM700 confocal microscope with a Plan Apochromat a Plan Neofluar 40×/1.30 NA oil lens and controlled using ZEN Black software. Representative images were prepared for publication using Fiji and Affinity Designer software.

For calculation of cytoneme density, optical sections of 400 µm² (Fig. 1C,D) or 80 µm² (Fig. 4D) in the *ptcGAL4* expression domain (labeled by GFP expression) were prepared and XY stacks were converted into XZ stacks using the 'reslice' function. Apical filopodia were quantified in the resulting image stacks by hand.

For quantification of split-GFP complementation (Fig. 1I,J), 8-bit images were acquired using separated channel tracks to reduce signal bleed-through. Identical settings were used for all conditions. Stacks were projected by maximum intensity projection and the wing pouch area was defined based on the Hoechst channel. The Hoechst images were then used to crop the images to remove non-specific GFP background. Using the Histogram function, lists of all pixel values were acquired. A threshold value was set at 75 (on 0–255-pixel value scale) based on analysis of the background signal in all three conditions. The calculated percentage of GFP positive surface was defined as the number of pixels above the threshold value divided by the total number of pixels.

For quantification of non-autonomous proliferation, the upper layers of image stacks where the PerM cells are located were projected using the maximum intensity projection function of Fiji to isolate images of the PerM of each disc. An 80 µm diameter circle was drawn in the region overlying the middle of the wing pouch (DP) where *ptcGAL4* is active. Using the cell counter function, nuclei inside the circle in the PM were then counted by hand.

For cryosection staining (Fig. EV1) PBT and related solutions were made as above using 0.3% Tween-20 instead of 0.2%. The anterior halves of *w^1118* third-instar larvae were isolated by dissection in ice-cold PBS and fixed in a solution of PBS containing 0.3% Tween-20 and 4% formaldehyde for 20 min. After 4 washes of 5 min each in PBT, anterior

halves were stained with DAPI (1 µg/ml, Invitrogen) and incubated in PBT with 15% sucrose then PBT with 30% sucrose until they reached equilibrium. Anterior halves were stripped of fat and other discs and then transferred into cryomolds (#22-363-553, Fisherbrand) filled with O.C.T. Compound (#4585, Fisher Healthcare). Discs were then isolated and oriented in O.C.T. followed by freezing in 99% ethanol in a dry ice bath. Samples were sectioned on a CryoStar NX70 cryostat (with the following parameters: module: −20 °C, blade: −16 °C and Cut thickness: 8 µm) and spread on glass slides. Slides were stained using an Epredia™ Shandon™ Sequenza™ Immunostaining Center Slide Rack (#73-310-017, Fisher Scientific) and phosphorylated Moesin (1/500, Cell Signalling Technology 3726S) or Armadillo (1/8000, AB_528089; Developmental Studies Hybridoma Bank) overnight at 4 °C. Secondary antibodies, imaging and processing were as described above.

## Statistical analysis

Quantification and graphs were prepared using Fiji for image analysis and GraphPad Prism10 for statistical analysis and graph building. Results are expressed as average ± SD. Statistical significance between various conditions was assessed by determining $p$ values (with a 95% confidence interval) using Prism software. Different tests were performed: unpaired $t$-test with Welch's correction or Mann–Whitney test (single comparison between two groups) and unpaired one-way Welch ANOVA to compare multiple groups, with a multiple comparisons test, corrected for sample size (Dunnett T3 when $n < 50$ and Games-Howell when $n > 50$). For parametric tests, a normal distribution across samples was assumed when $n \geq 30$. When $n < 30$, the normality of the data was formally assessed using a D'Agostino-Pearson test. Non-parametric tests were applied if these conditions were not met.

## Data availability

The source data for images produced in this study are available in BioImage Archive: https://doi.org/10.6019/S-BIAD1490.

The source data of this paper are collected in the following database record: biostudies:S-SCDT-10_1038-S44318-025-00401-8.

## Peer review information

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

## Acknowledgements

This work has been supported by CIHR (PJT-162109 to DH and SC) and NSERC Discovery Grants (RGPIN to SC and MJS). BR held a doctoral scholarship from the Institute for Research in Immunology and Cancer and from University of Montreal's Molecular Biology Program. CDJ held a scholarship from Wallonie-Bruxelles International and a doctoral scholarship from the Institute for Research in Immunology and Cancer and from University of Montreal's Molecular Biology Program. RS was supported by scholarships from the Cole Foundation and the Fonds de recherche du Québec – Nature et Technologies (FRQNT). MJ held a doctoral scholarship from the Institut de recherches cliniques de Montréal Foundation and from University of Montreal's Molecular Biology Program. F-CT and PB are members of the CNRS consortium AQV. MJS holds a Canada Research Chair (CRC) in Cancer Signalling and Structural Biology. F-CT and PB are members of the Labex Cell(n)Scale (ANR-11-LABX0038) and Paris Sciences et Lettres (ANR-10-IDEX-0001-02). The cDNAs of the I-BAR domains of MIM and IRSp53 were a kind gift from Pekka Lappalainen (University of Helsinki). The authors greatly acknowledge the Cell and Tissue Imaging core facility (PICT IBiSA), Institut Curie, member of the French National Research Infrastructure France-BioImaging (ANR10-INBS-04), the Electron Imaging Facility, Faculty of Dental Medicine, Université de Montréal; the IRIC Bio-imaging Core Facility and IRCM Microscopy and Imaging platform; and Dr. Marine Lacomme (IRCM) for assistance with cryosectioning.

## Author contributions

**Basile Rambaud**: Conceptualization; Data curation; Software; Formal analysis; Supervision; Investigation; Visualization; Methodology; Writing—original draft; Writing—review and editing. **Mathieu Joseph**: Conceptualization; Data curation; Formal analysis; Supervision; Investigation; Visualization; Methodology; Writing—original draft; Writing—review and editing. **Feng-Ching Tsai**: Conceptualization; Data curation; Validation; Investigation; Visualization; Methodology; Writing—original draft; Writing—review and editing. **Camille De Jamblinne**: Conceptualization; Formal analysis; Validation; Visualization. **Regina Strakhova**: Formal analysis; Investigation; Methodology. **Emmanuelle Del Guidice**: Investigation. **Renata Sabelli**: Investigation. **Matthew J Smith**: Conceptualization; Data curation; Formal analysis; Funding acquisition; Investigation; Writing—original draft; Writing—review and editing. **Patricia Bassereau**: Conceptualization; Resources; Data curation; Formal analysis; Supervision; Funding acquisition; Validation; Investigation; Visualization; Writing—original draft; Project administration; Writing—review and editing. **David R Hipfner**: Conceptualization; Resources; Data curation; Formal analysis; Supervision; Funding acquisition; Validation; Investigation; Visualization; Writing—original draft; Project administration; Writing—review and editing. **Sébastien Carréno**: Conceptualization; Data curation; Formal analysis; Supervision; Funding acquisition; Investigation; Visualization; Writing—original draft; Project administration; Writing—review and editing.

Source data underlying figure panels in this paper may have individual authorship assigned. Where available, figure panel/source data authorship is listed in the following database record: biostudies:S-SCDT-10_1038-S44318-025-00401-8.

## Disclosure and competing interests statement

The authors declare no competing interests.

# Expanded View Figures

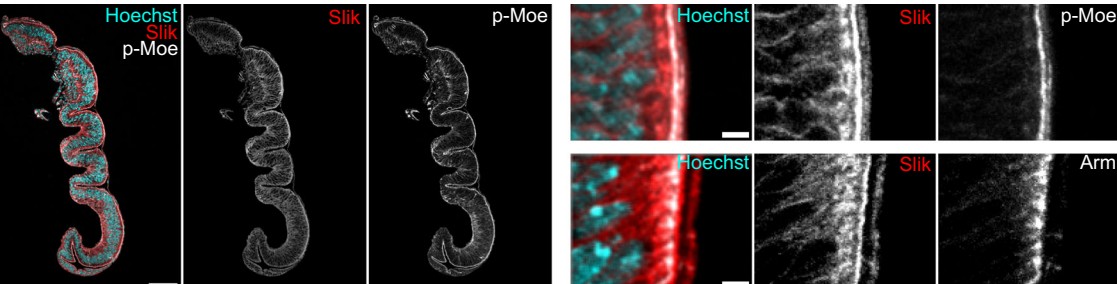

**Figure EV1.  Slik localizes to the apical-most free membrane in wild-type wing disc DP cells.**

8 μm transverse section of $w^{1118}$ wing disc prepared using cryosectioning protocol and immunostained for Slik (in red), phospho-Moesin (in white, left and top right) or Armadillo (β-catenin; in white, bottom right). Nuclei are stained with Hoechst (cyan). Scale bar = 50 μm (left) or 3 μm (right).

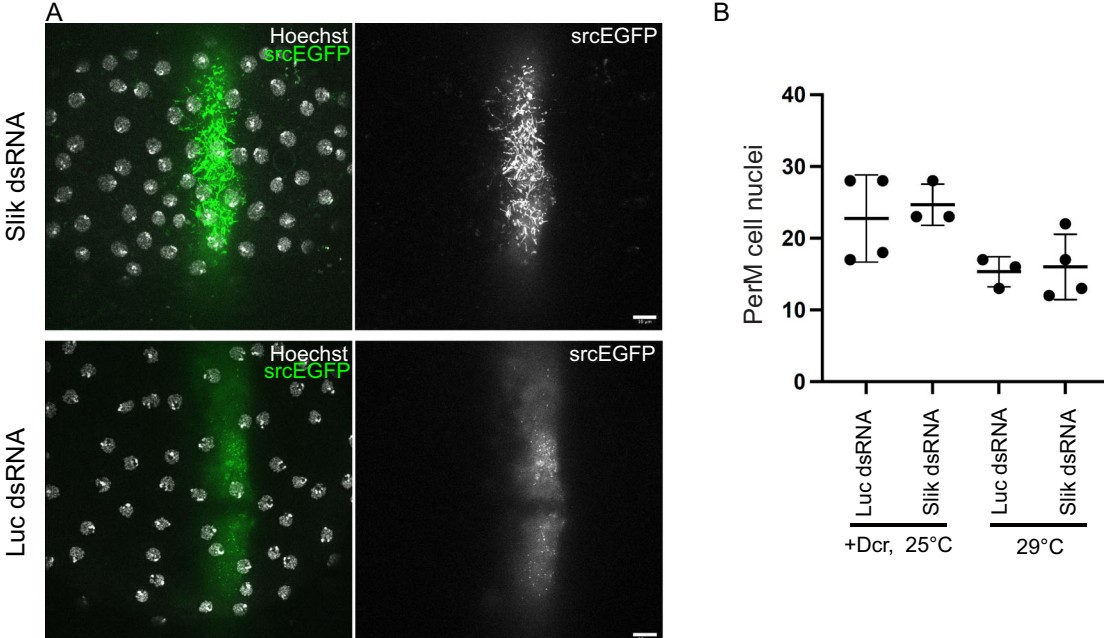

**Figure EV2.  Loss of Slik function induces nonfunctional filopodia.**

(**A**) Projection of confocal sections through the PerM and apical-most region of the DP of discs expressing dsRNA targeting Slik (top) or Luciferase (control, bottom), together with the membrane marker srcEGFP (green), under the control of *ptc*-GAL4. Nuclei were stained with Hoechst (white). Scale bar = 10 μm. Slik depletion in srcEGFP-expressing DP cells caused the appearance of apical filopodia. (**B**) Quantification of the number of PerM cell nuclei in an 80 μm diameter circle above the DP expression domain of dsRNA targeting Slik or Luciferase, under the control of *ptc*-GAL4. The experiment was performed in two different conditions: with co-expression of Dicer (Dcr) at 25 °C (left), or at higher temperature (29 °C) without Dcr (right). Analysis was performed on (from left to right) 4, 3, 3, and 4 discs. Error bars indicate mean ± s.d. The filopodia formed upon Slik depletion did not support proliferative signaling.

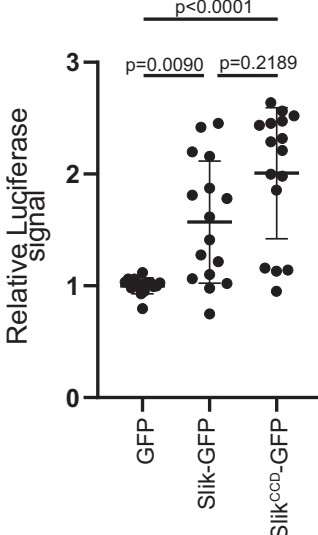

**Figure EV3. Slik^{CCD}-dependent cytonemes are competent for cell–cell communication.**

Co-culture experiment between donor cells expressing GFP, Slik-GFP or Slik^{CCD}-GFP and Hh and acceptor cells expressing Ci and Ptc-luciferase. GFP and Slik-GFP conditions were already shown in Fig. 2F. $n = 16$ from 4 independent experiments, each performed in quadriplicate. *P* values were calculated using Kruskal–Wallis test with Dunn's test for multiple comparisons. Source data are available online for this figure.

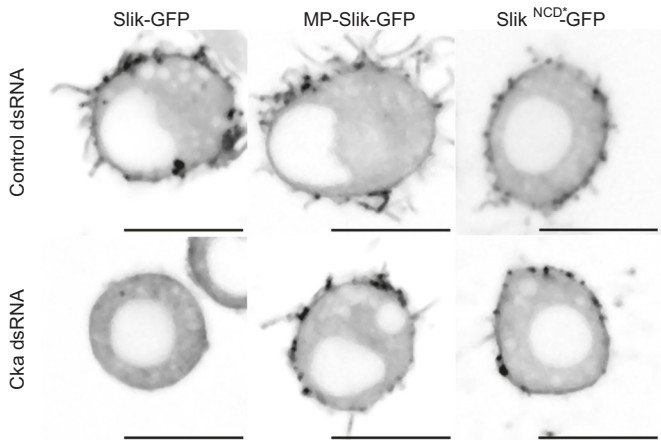

**Figure EV4.  MP-Slik-GFP and Slik<sup>NCD*</sup>-GFP are still localized at the plasma membrane after Cka depletion.**

Confocal microscopy images of cells expressing Slik-GFP, MP-Slik-GFP, or Slik<sup>NCD*</sup>-GFP after treatment with dsRNA targeting Gal4 (Control, top row) or Cka (bottom row). Scale bars = 10 µm.

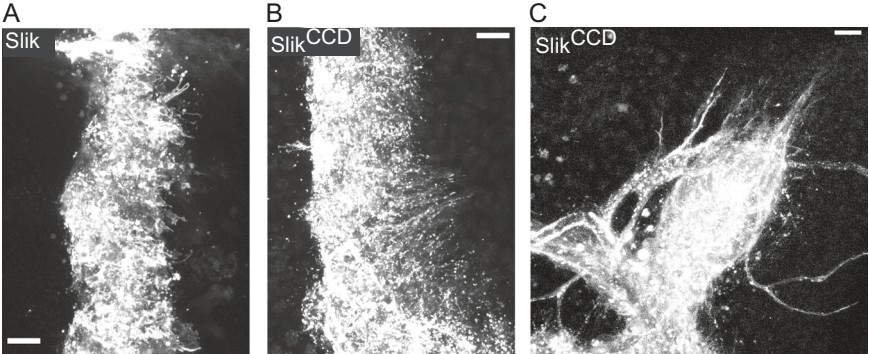

**Figure EV5. Slik-GFP and Slik^CCD-GFP localize to cytonemes on the basal DP surface and on the air sac primordium.**

(A, B) Projection of confocal sections through the basal region of the DP in discs expressing Slik-GFP (A) or Slik^CCD-GFP (B) under the control of *ptc*-GAL4. Slik localized to cytonemes emanating from the basal side of DP cells. Scale bars = 10 μm. (C) Projection of confocal sections through a wing sac primordium expressing Slik^CCD-mCherry under the control of *btl.S*-GAL4. Slik localized to cytonemes emanating from the air sac primordium. Scale bars = 10 μm.

