## [Peer Review File · The EMBO Journal]

Slik sculpts the plasma membrane into cytonemes to control cell-cell communication

Sébastien Carreno, Basile Rambaud, Mathieu Joseph, Feng-Ching Tsai, Camille De Jamblinne, Regina Strakhova, Emmanuelle Del Guidice, Renata Sabelli, Matthew Smith, Patricia Bassereau, and David Hipfner

Corresponding authors: Sébastien Carreno (sebastien.carreno@umontreal.ca) , David Hipfner (david.hipfner@ircm.qc.ca)

Review Timeline:

Submission Date:	22nd Apr 24
Editorial Decision:	27th May 24
Revision Received:	15th Nov 24
Editorial Decision:	17th Dec 24
Revision Received:	8th Feb 25
Accepted:	19th Feb 25

Editor: Ieva Gailite

Transaction Report:

Dear Dr. Carreno,

Thank you for submitting your manuscript for consideration by the EMBO Journal. We have now received a full set of reviewer reports, which are included below for your information.

As you will see from the reports, the reviewers find the study of interest, while also pointing out several controls and further experiments that would be required to strengthen the conclusiveness of the study. Based on the interest expressed in the reports, I would like to invite you to address the concerns raised by the referees in a revised manuscript. I think it would be useful to discuss the revision in more detail via email or phone/videoconferencing - please let me know which option you prefer.

We generally allow three months as standard revision time. As a matter of policy, competing manuscripts published during this period will not negatively impact on our assessment of the conceptual advance presented by your study. However, please contact me as soon as possible upon publication of any related work to discuss the appropriate course of action. Should you foresee a problem in meeting this three-month deadline, please contact us to arrange an extension.

When preparing your letter of response to the referees' comments, please bear in mind that this will form part of the Review Process File and will therefore be available online to the community. For more details on our Transparent Editorial Process, please visit our website: <https://www.embopress.org/page/journal/14602075/authorguide#transparentprocess>. Please also see the attached instructions for further guidelines on preparation of the revised manuscript.

Please feel free to contact me if you have any further questions regarding the revision. Thank you for the opportunity to consider your work for publication. I look forward to discussing your revision.

With best wishes,

leva

leva Gailite, PhD
Senior Scientific Editor
The EMBO Journal
Meyerhofstrasse 1
D-69117 Heidelberg
Tel: +4962218891309
i.gailite@embojournal.org

We realize that it is difficult to revise to a specific deadline. In the interest of protecting the conceptual advance provided by the work, we recommend a revision within 3 months (25th Aug 2024). Please discuss the revision progress ahead of this time with the editor if you require more time to complete the revisions.

Referee #1:

This paper describes a potentially novel membrane-sculpting function of the coiled-coil domain of the Slik protein in *Drosophila*. The membrane sculpting would be similar to that induced by the BAR domain proteins. Several points need to be addressed.

1. Cytoneme vs. filopodia issue. The cytoneme is controversial and would not be different from filopodia. Many filopodia do not attach to the substratum because they are considered to be probing the surrounding cells. If the cytoneme needs to be differentiated from filopodia, then the exclusion of filopodial proteins, including I-BAR domain proteins, would be examined.
2. Figure 1F, The contact analysis of the GFP1-10 to GFP11 binding is interesting. The maturation for giving fluorescence by the complex formation takes hours. However, the contact might not be so stable. Then, the contact would be something induced by the binding of GFP1-10 to GFP11. Some control experiments would be required, including the endogenous protein stain at the adhesion sites.
3. In Figure 2, the frequency of the cytoneme vs. filopodia would be examined as per the authors' definition by using Slik localization and the I-BAR protein localization.
4. Depletion of Slik from Fig 2F and Cka dsRNA depletion in Fig 6 A&C showed a decrease in the number of cytonemes but did not affect the length of cytonemes. What are the proposed reasons for these observations?
5. Figs. 3A and 3B. It would be helpful to observe the phosphorylation of Moesin by western blotting and relate it to the cytoneme biogenesis.
6. Figure 4D, they mentioned that large increase in the number of apical cytonemes was observed in Slik CCD expressing DP cells compared to controls. But what about comparing Silk CCD and Slik Full Length (FL)? Is there any difference observed between these two in apical cytoneme generation?
7. In Figure 4G, they mentioned that lateral spikes were formed by an accumulation of Slik CCD -GFP, but unlike the body of cytonemes, they were devoid of F-actin. But why were they devoid of F-actin? If the Silk CCD was devoid of F-actin, how could it contribute to cytoneme elongation without the mechanical force from F-actin? It would be similar to I-BAR.
8. Figure 5. The BAR domains form dimer and the prediction of the dimer structure would be required.
9. The membrane binding sites on the Slik structure would be modeled with mutations. Is it possible to show an electrostatic density map of the CCD to show the probable binding sites of the domain to the plasma membrane?
10. In Figure 5D, how many GUVs have they observed to present the data? What is the percent of outward tubules compared to inward tubules on the GUV? Also, can they compare the GUV images of MIM I-BAR and IRSp53 I-BAR with Silk CCD and then compare various differences in the features of inward and outward tubules between these three?
11. Phosphorylation would be reasonable for suppressing membrane binding and cytoneme formation because the negative charges introduced by the phosphorylation counteract the positive charge for membrane binding. However, this does not happen on the CCD. Then, how does the negative charge introduced by phosphorylation affect the CCD-to-membrane interaction? One candidate is the autoinhibitory interaction. Another is the phosphorylated region is in a structural unit with CCD. Structural prediction will be required.
12. In Fig 6B, they said that, in contrast to wild-type Slik, Slik NCD* maintained its association to the plasma membrane even when dSTRIPAK is inactivated. But what is the quantification to claim this? If already they have quantification, interpretation of the data is required.
13. After Fig 6D, they have mentioned that time-lapse movies showed a decrease in cytoneme dynamics in strip/cka-depleted cells. But the data was not shown in the manuscript. Can they add this data?
14. If the phosphorylation site is not conserved, then how can this mechanism be important?
15. The author's meticulous consideration of various factors, including incorporating statistical tests such as Welch correction, Dunnett T3, and Games-Howell for addressing issues of unequal variances and sample sizes, is commendable. However, I would recommend further attention to conducting a normality test prior to selecting the appropriate statistical test. This step is crucial as deviations from normality in the data can compromise the accuracy of parametric tests, potentially resulting in inflated

Type I error rates or biased estimates. Would it be more appropriate to utilize a paired t-test for analyzing Figure 4H, given its depiction of time-lapse cytoneme growth?

Minor points:

In Figure 1E, the extension and contraction of cell protrusions were observed. How was the focus determined? 3D observation would be required. It is important to quantify this. For instance, how much change in lengths has occurred for the protrusions?

Figure 3A would be better to have the statistical significance between Slik and GFP groups.

Figure 1C graph label has unnecessary parentheses.

In Figure 3C, the arrangement of the pictures and graph elements should be in the same order.

In Figures 2F, 3, and 4B, the lower error bars were missing.

The number of cells that were measured in each graph should be shown.

The origin of S2 cells should be mentioned.

The additional data that show protein expression levels (for ex, Western Blot) should be provided.

The legend for Fig5E should be for Fig5D.

Referee #2:

This is a scholarly, well-conceived, and technically impressive study of the important but poorly understood process that is responsible for the initiation and elongation of filopodia. This study focuses on cytoneme signaling filopodia in the *Drosophila* wing disc and S2 cultured cells and provides evidence for a role of the Slik protein. My suggestions are minor:

Fig 1A Labeling is confusing - I am guessing that LifeActRFP is red in left panels and white in right panels? Perhaps add red LifeAct label to left panel and white LifeAct label to right panel? Or keep RFP signal consistent/red? It might also help to label/indicate DP and PerM. Is ptc expression only in DP in the imaged portion of the disc, accounting for absence of fluorescence in PerM in upper right panel?

Fig 1B Confusing labels as in 1A. Higher magnification views of the regions of interest might be helpful.

Studies of oogenesis have identified ectopic artifacts upon LifeAct overexpression. Do these discs with LifeAct overexpression make normal adult wings?

Fig 1G Please explain contrasting densities in the upper panels

Fig 2A For the uninitiated, please explain "stained for F-actin", difference between and reason for use of MAX projection and 3D, basic methodology and utility for use of CLSEM

Fig 2F How were unattached cytonemes defined experimentally?

Fig 4C Hoechst staining may be detecting wolbachia? What is the marginal zone?

Fig 4E Is over-proliferation of PerM cells associated with Dpp signal transduction?

Fig. 4F Please identify lateral spikes.

Fig 4G Images show that the F-actin staining was insufficiently sensitive to detect presence of F-actin, but does it show that the spikes are devoid of F-actin? Does the staining detect a single actin filament?

Fig 5A "revealed" or "predicted" this structure?

Fig 5C How was the purity of Slik[CCD] determined?

Fig 5D Correct legend: change (E) to (D)

Referee #3:

In this work, Rambaud et al. discover that the *Drosophila* Ser/Thr kinase Slik plays a crucial role in regulating membrane extension formation, explaining its previously known role in distant cell proliferation. Using a combination of complementary approaches *in vitro* in S2 cells and *in vivo* in wing discs, the authors identify a domain in Slik protein that directly shapes the membrane into filopodia, initiating cytoneme formation. This domain does not require Slik kinase activity. Furthermore, membrane elongation can be prevented by phosphorylation of its target Moesin. Finally, they found that the dSTRIPAK complex, the first family of kinases to directly shape the membrane, also plays an essential role in membrane sculpting function by regulating the association of Slik with the plasma membrane.

Although the authors have not yet identified the mechanisms that regulate this membrane-forming induction, the evidence that activity of the C-terminal coiled-coil domain of Slik acts as a novel regulator of the biogenesis of membranes to control the proliferation of cells at a distance is very strong. In addition, the paper contains very interesting findings: unlike other I-BAR domain proteins, SlikCCD binds to Ptdins(4)P and is the first membrane sculpting domain capable of inducing both negative and positive curvatures of Giant Unilamellar Vesicles *in vitro*. All the data are very well presented and the manuscript is well written. My major problem with this manuscript is why the apical extensions are considered to be cytonemes.

In summary, I feel that this paper merits publication after some revision.

Major points

1) My major concern with this manuscript is if the apical membrane extension are proper cytonemes. There is no evidence that these membrane protrusions perform signaling in a physiological situation. It would be important to see these apical membrane protrusions by EM.

Most publications on cytonemes implicated in cell-cell signaling involving different pathways and tissue communication in the wing disc have been described at the basolateral and basal side of the disc epithelium. If Slik has general membrane sculpting functions to control biogenesis of various specialized membrane protrusions it would be important to show or at least discuss if Slik could have also sculpting membranes function basally.

Here, the function of Slik in the membrane biogenesis has only been analyzed in the apical plasma membrane of the wing disc. A possible reason for this could be that Slik protein localizes only apically; if this is the case, this has to be well explained in the manuscript.

2) Although the effect of Slik-GFP overexpression to analyze membrane extensions in the wing disc is shown in Fig 1 and Fig 4 C, experiments of Slit loss of function have been performed in S2 cells but not in the wing disc. I have only found one experiment that has been carried out in the wing disc using Strip dsRNA but in this case SlikGFP is overexpressed simultaneously (Fig 6D). I consider important to show the effect of the absence of Slik in the wing imaginal disc. If confocal images do not provide enough resolution to visualize the microvilli, EM images could overcome this problem.

3) While Slik seems to promote the delivery of growth signals from DP to PerM cells through an increase in the number and length of apical cytonemes, the signals involved in this transluminal cytoneme-mediated communication have not been identified in this work. Besides, the described apical effect of ectopic DP over the PM could be simply a consequence of the overexpression. The GRAP signal shown in Fig 2 G occurs only under Slik overexpression and not, or barely, in normal conditions.

Minor points:

1) In several places along the manuscript some data are presented as "data are not shown". The data should be shown (at least as supplementary material).

As examples:

"In time-lapse movies, there was also a decrease in cytoneme dynamics in strip/cka-depleted cells (not shown)".

"Although we have obtained experimental evidence indicating that SlikCCD directly binds to actin filaments (data not shown)"....

"We found that SlikCCD binds to Ptdins(4)P rather than Ptdins(4,5)P2 like I-BAR domains do, and SlikCCD sculpts GUVs containing Ptdins(4)P but not Ptdins(4,5)P2 (data not shown)".

2) Figure 2 C, E. describe the averaging cytoneme diameter as 172 nm and 181 nm. The authors mention that this size coincides with the averaging diameter of cytonemes found by Tom Kornberg (Kornberg and Roy, 2014). I find more appropriated to also cite the reference Wood et al., 2021, since this article defines the cytoneme diameter by electron microscopy.

3) In Fig 2F, while Slik overexpression increases the length of membrane extensions in S2 cells, its depletion does not result in their shortening; instead, cytonemes appear longer than in control cells. An explanation or a comment on this fact is needed.

4) Fig. 4C. I have not been able to see the promotion of cytoneme formation in DP cells expressing SlikKD-GFP.

5) Fig 6F and Fig 6 G are not quoted in the main manuscript text.

6) It was a surprise to me to find out that the title of this manuscript was based on the results of the last figure.

Point-by-point response to the remaining reviewer comments:

Referee #1:

This paper describes a potentially novel membrane-sculpting function of the coiled-coil domain of the Slik protein in *Drosophila*. The membrane sculpting would be similar to that induced by the BAR domain proteins. Several points need to be addressed.

1. Cytoneme vs. filopodia issue. The cytoneme is controversial and would not be different from filopodia. Many filopodia do not attach to the substratum because they are considered to be probing the surrounding cells. If the cytoneme needs to be differentiated from filopodia, then the exclusion of filopodial proteins, including I-BAR domain proteins, would be examined.

Unfortunately, there are currently no specific markers to reliably distinguish cytonemes from non-signaling filopodia. However, to address this concern, we performed a series of new experiments to test whether Slik-induced protrusions can facilitate the delivery of morphogens to neighboring cells in culture. We conducted co-culture experiments using Hedgehog-expressing donor cells and receiving cells with a Hedgehog-responsive promoter. Consistent with the hypothesis that Slik-induced protrusions are bona fide cytonemes, we found that the expression of Slik or SlikCCD in Hedgehog donor cells led to an increase in Hedgehog-responsive promoter activity in the receiving cells. These results are now presented in Fig 2E-F, EV3 and 6B.

2. Figure 1F, The contact analysis of the GFP1-10 to GFP11 binding is interesting. The maturation for giving fluorescence by the complex formation takes hours. However, the contact might not be so stable. Then, the contact would be something induced by the binding of GFP1-10 to GFP11. Some control experiments would be required, including the endogenous protein stain at the adhesion sites.

The apical cytonemes that we describe in discs are not well preserved by fixation, limiting our ability to conduct immunofluorescence studies on them. As an alternative way to confirm that there is contact between DP and PerM cells that is not dependent on the interaction between GFP1-10 and GFP11, we turned to live imaging of discs expressing just Slik-GFP or SlikCCD-GFP in the DP. In such discs, while many Slik/SlikCCD-containing filopodia are dynamically extending and retracting, we also see a population of stable cytonemes connecting DP cells to PerM cells. Many of these cytonemes persist for the full duration of the timelapse experiments, up to 90 minutes. This suggests that there is stable contact between the two layers mediated by these cytonemes, that occurs independent of GFP complementation. We added a movie (Movie 1) to illustrate this point.

3. In Figure 2, the frequency of the cytoneme vs. filopodia would be examined as per the authors' definition by using Slik localization and the I-BAR protein localization.

As mentioned in our response to point 1, there are currently no specific markers that reliably distinguish cytonemes from non-signaling filopodia. Consequently, using Slik or I-BAR protein localization alone does not provide a definitive differentiation between these structures.

4. Depletion of Slik from Fig 2F and Cka dsRNA depletion in Fig 6 A&C showed a decrease in the number of cytonemes but did not affect the length of cytonemes. What are the proposed reasons for these observations?

While the dSTRIPAK-Slik signaling axis regulates cytoneme biogenesis, additional signaling pathways may also contribute to the formation of these structures. Thus, the reduction in cytoneme numbers observed upon Slik or Cka depletion may be compensated by other proteins, such as other I-BAR proteins, that help maintain cytoneme length. In addition, in absence of Cka or Slik, Moesin is not activated and therefore cannot counteract cytoneme elongation. We have now clarify this in the discussion section by adding: ***'In S2 cells without Slik, its CCD cannot tubulate the plasma membrane into nascent cytonemes, leading to fewer cytonemes per cell. However, in the absence of Slik kinase activity, Moesin is not activated, failing to regulate cytoneme length, which may be sustained by other I-BAR proteins.'***

5. Figs. 3A and 3B. It would be helpful to observe the phosphorylation of Moesin by western blotting and relate it to the cytoneme biogenesis.

Fig. 3A: We have previously demonstrated by western blotting that Slik overexpression increases Moesin phosphorylation in S2 cells. We thus decided that repeating this experiment was not essential, but we have cited our prior publications that provide this evidence.

Fig. 3B: Concanavalin A was shown to stiffen the cortex from the extracellular side, independently of Moesin activity. We have added this information to the text for clarity.

6. Figure 4D, they mentioned that large increase in the number of apical cytonemes was observed in Slik CCD expressing DP cells compared to controls. But what about comparing Silk CCD and Slik Full Length (FL)? Is there any difference observed between these two in apical cytoneme generation?

As mentioned in the manuscript, there are qualitative differences in the effects of Slik and Slik^{CCD} on apical membrane morphology in discs. Whereas Slik^{CCD} induces only the formation of apical cytonemes, Slik additionally causes the widespread appearance of larger apical membrane blebs. This complicates a direct quantitative comparison of their effects on cytoneme formation. Furthermore, although both are driven by the same driver, we have no good way of precisely comparing the levels of Slik and Slik^{CCD} protein in these experiments, making it difficult to make a biologically meaningful comparison.

7. In Figure 4G, they mentioned that lateral spikes were formed by an accumulation of Slik CCD - GFP, but unlike the body of cytonemes, they were devoid of F-actin. But why were they devoid

of F-actin? If the Silk CCD was devoid of F-actin, how could it contribute to cytoneme elongation without the mechanical force from F-actin? It would be similar to I-BAR.

As noted by Reviewer 1, we cannot completely rule out the presence of some actin filaments within the Slik^{CCD} lateral spikes that might not be detectable by phalloidin staining. We have revised the text to describe these spikes as being **“devoid of detectable F-actin.”** Additionally, as mentioned in the Discussion, we hypothesize that these Slik^{CCD} spikes **“may represent a transient intermediate stage in cytoneme biogenesis, possibly marking the initial step of plasma membrane tubulation into cytonemes.”** To further clarify this, we have included a model in the first paragraph of the Discussion that outlines how the STRIPAK-Slik signaling axis promotes cytoneme formation and elongation.

8. Figure 5. The BAR domains form dimer and the prediction of the dimer structure would be required.

We conducted additional experiments to determine whether Slik^{CCD} forms dimers *in vitro* by analyzing the purified domain using gel filtration. Our results showed that Slik^{CCD} eluted with a molecular weight consistent with dimerization, which is now presented in Fig 5C. Although we obtained experimental evidence establishing Slik^{CCD} dimer formation, AlphaFold Multimer failed to predict the dimer structure of Slik^{CCD}.

9. The membrane binding sites on the Slik structure would be modeled with mutations. Is it possible to show an electrostatic density map of the CCD to show the probable binding sites of the domain to the plasma membrane?

The electrostatic density maps of BAR domain dimers can predict their probable binding sites on the plasma membrane. When aligned with the positive or negative curvatures of these dimers, this approach can help model how BAR domains tubulate membranes. However, without a predicted dimer structure for Slik^{CCD}, we cannot perform a similar analysis. We did generate an electrostatic density map for the monomeric form of Slik^{CCD} (See Figure 1 of this rebuttal), but it did not reveal any clear membrane-binding sites associated with positive or negative curvature. Therefore, we decided not to include this analysis in the manuscript.

Figure for reviewers removed.

10. In Figure 5D, how many GUVs have they observed to present the data? What is the percent of outward tubules compared to inward tubules on the GUV? Also, can they compare the GUV images of MIM I-BAR and IRSp53 I-BAR with Silk CCD and then compare various differences in the features of inward and outward tubules between these three?

The statistics from independent experiments with GUVs and Slik^{CCD} are now presented in Figure 5F. The effects of MIM I-BAR and IRSp53 I-BAR on GUVs, using the same experimental procedures, have been previously reported (PMID: 19150238; PMID: 38724689; PMID: 36240267). As mentioned in the manuscript, outward tubulation of GUVs was never observed in those studies. While these published findings offer valuable context, we have concentrated our analysis on the specific effects of Slik^{CCD} to maintain a clear focus on the central questions of our study.

11. Phosphorylation would be reasonable for suppressing membrane binding and cytoneme formation because the negative charges introduced by the phosphorylation counteract the positive charge for membrane binding. However, this does not happen on the CCD. Then, how does the negative charge introduced by phosphorylation affect the CCD-to-membrane

interaction? One candidate is the autoinhibitory interaction. Another is the phosphorylated region is in a structural unit with CCD. Structural prediction will be required.

We thank the reviewer for this insightful suggestion. We have indeed identified some residues whose phosphorylation may regulate autoinhibitory interactions and control tubulation activity. While these preliminary findings are promising, they require extensive experimental validation, which will be best suited for a follow-up study dedicated to exploring these molecular mechanisms in detail.

12. In Fig 6B, they said that, in contrast to wild-type Slik, Slik NCD* maintained its association to the plasma membrane even when dSTRIPAK is inactivated. But what is the quantification to claim this? If already they have quantification, interpretation of the data is required.

The quantification for this observation was previously reported in De Jamblinne et al. (2020) - PMID: 32960945. We recognize that our original text may have been misleading, and we have revised it to: *"To examine whether dSTRIPAK modulates cytoneme biogenesis by regulating Slik's association with the plasma membrane, we expressed a non-phosphorylatable mutant of Slik, in which the 17 Ser/Thr residues of SlikNCD were replaced by Ala (SlikNCD-GFP). In contrast to wild-type Slik, **we previously established that SlikNCD**, maintained its association with the plasma membrane even when dSTRIPAK was inactivated (De Jamblinne et al., 2020)."*

13. After Fig 6D, they have mentioned that time-lapse movies showed a decrease in cytoneme dynamics in strip/cka-depleted cells. But the data was not shown in the manuscript. Can they add this data?

We have now included the movies in the manuscript.

14. if the phosphorylation site is not conserved, then how can this mechanism be important?

We have not specifically tested for conservation of phosphorylation sites between Slik and its two human orthologs, SLK and LOK. However, our previous findings (De Jamblinne et al, 2020 - PMID: 32960945) suggest that it is the global phosphorylation state of Slik that modulates its association with the plasma membrane. Both SLK and LOK are phosphorylated at multiple Ser/Thr residues in their non-conserved regions or their CCDs. While this model of membrane association still needs formal validation in mammals, it is plausible that the mechanism by which Slik kinases interact with the plasma membrane is conserved across evolution.

LOK : <https://www.phosphosite.org/proteinAction.action?id=761&showAllSites=true>

SLK : <https://www.phosphosite.org/proteinAction.action?id=2519&showAllSites=true>

15. The author's meticulous consideration of various factors, including incorporating statistical tests such as Welch correction, Dunnett T3, and Games-Howell for addressing issues of unequal variances and sample sizes, is commendable. However, I would recommend further attention to conducting a normality test prior to selecting the appropriate statistical test. This step is crucial

as deviations from normality in the data can compromise the accuracy of parametric tests, potentially resulting in inflated Type I error rates or biased estimates. Would it be more appropriate to utilize a paired t-test for analyzing Figure 4H, given its depiction of time-lapse cytoneme growth?

For experiments with $n \geq 30$, we assumed normality based on the Central Limit Theorem, which states that the sampling distribution of the mean approximates a normal distribution with a sufficiently large sample size. For experiments with $n < 30$, we formally tested data normality using the D'Agostino-Pearson test. In the case of Figure 4H, a paired t-test was conducted, as it was more appropriate for the data, and normality was confirmed beforehand using the D'Agostino-Pearson test. This is now explained in the Material and Methods section.

Minor points:

In Figure 1E, the extension and contraction of cell protrusions were observed. How was the focus determined? 3D observation would be required. It is important to quantify this. For instance, how much change in lengths has occurred for the protrusions?

The images presented are maximum projections of confocal stacks capturing regions from above to below the cytoneme protrusions. Notably, the cytonemes extend into the disc lumen, which is relatively narrow. Those extending dynamically over longer distances typically project laterally within the lumen and are contained within 5 to 6 image slices spanning a maximum of 4 microns.

The timelapse imaging was done by confocal microscopy with the goal of capturing high Z-resolution. Each individual image stack was thus relatively slow to acquire, and we were only able to take an image every two minutes. This is too slow to allow meaningful measurement of rates of change in protrusion length. However, the effect is clearly observable even without precise quantification. To illustrate this more effectively, we have included timelapse movies in the manuscript (movie 1).

Figure 3A would be better to have the statistical significance between Slik and GFP groups.

Figure 1C graph label has unnecessary parentheses.

In Figure 3C, the arrangement of the pictures and graph elements should be in the same order.

In Figures 2F, 3, and 4B, the lower error bars were missing.

The number of cells that were measured in each graph should be shown.

The origin of S2 cells should be mentioned.

The additional data that show protein expression levels (for ex, Western Blot) should be provided.

The legend for Fig5E should be for Fig5D.

This has been corrected in the revised manuscript.

Referee #2:

This is a scholarly, well-conceived, and technically impressive study of the important but poorly understood process that is responsible for the initiation and elongation of filopodia. This study focuses on cytoneme signaling filopodia in the *Drosophila* wing disc and S2 cultured cells and provides evidence for a role of the Slik protein. My suggestions are minor:

Fig 1A Labeling is confusing - I am guessing that LifeActRFP is red in left panels and white in right panels? Perhaps add red LifeAct label to left panel and white LifeAct label to right panel? Or keep RFP signal consistent/red? It might also help to label/indicate DP and PerM. Is *ptc* expression only in DP in the imaged portion of the disc, accounting for absence of fluorescence in PerM in upper right panel? Fig 1B Confusing labels as in 1A. Higher magnification views of the regions of interest might be helpful. Studies of oogenesis have identified ectopic artifacts upon LifeAct overexpression. Do these discs with LifeAct overexpression make normal adult wings?

The labelling of Fig 1 has been clarified. We don't observe much effect in the adult wing of *ptc*-GAL4-driven LifeAct-RFP expression. In fact, we keep this as a recombinant stock with no obvious deleterious effects.

Fig 1G Please explain contrasting densities in the upper panels

We assume this comment is referring to contrasting density of nuclei in the images? In the left and middle panels, the discs are wild-type with respect to Slik expression, and both show a normal number of PerM nuclei. The disc on the right is overexpressing Slik specifically in the DP wing pouch under the control of *nubbin*-GAL4, and this drives dramatic overproliferation of the overlying PerM cells leading to a large increase in nuclear density. We have added a comment on this to the manuscript text.

Fig 2A For the uninitiated, please explain "stained for F-actin", difference between and reason for us of MAX projection and 3D, basic methodology and utility for use of CLSEM

Fig 2F How were unattached cytonemes defined experimentally?

This has been better explained in the text.

Fig 4C Hoechst staining may be detecting *Wolbachia*? What is the marginal zone?

We don't think there is Hoechst staining of *Wolbachia* in the image. Rather the Hoechst channel in the image was a bit underexposed, and some background appeared when we increased the

brightness to make the nuclei more visible. We went back to the original and made slight adjustments to the image brightness to reduce the background.

The marginal zone is a region of the apical membrane in *Drosophila* epithelia just apical to the adherens junctions (as opposed to the “free apical membrane”). This may not be a widely recognized term among non-*Drosophila* epithelial biologists. We adjusted the text for clarity: ***“Slik^{CCD}-GFP localized to dynamic cytonemes projecting from the apical surface of DP cells, most prominently in the marginal zone where neighbouring cells make contact.”***

Fig 4E Is over-proliferation of PerM cells associated with Dpp signal transduction?

Given previous evidence that Dpp can mediate signal transduction in the wing disc, we tested by genetic interactions for the involvement of this pathway in the proliferation phenotype induced by Slik. Dpp signaling is not a major contributor to Slik-mediated transduction, as there was little or no effect on the non-autonomous proliferation phenotype in discs heterozygous for the genes encoding the Dpp receptor Tkv or its downstream target, the transcriptional regulator Mad ((See Figure 2 of this rebuttal)

Figure for reviewers removed.

In contrast, we do see strong dosage-sensitive suppression of Slik overexpression phenotypes in a heterozygous background for components of the Ras-MAPK signaling pathway, and we believe we have identified the responsible ligand. While this is a promising direction, it is still a work in progress. Given the primary focus of this manuscript on Slik’s role in membrane shaping, we plan to address the mechanistic details of this pathway interaction in future studies.

Fig. 4F Please identify lateral spikes.

The lateral spikes have been identified.

Fig 4G Images show that the F-actin staining was insufficiently sensitive to detect presence of F-actin, but does it show that the spikes are devoid of F-actin? Does the staining detect a single actin filament?

Indeed, we cannot completely rule out the presence of some actin filaments within the **Slik^{CCD} lateral spikes that might not be detectable by phalloidin staining**. We have revised the text to describe these spikes as being “devoid of detectable F-actin.”

Fig 5A "revealed" or "predicted" this structure?

Fig 5C How was the purity of Slik[CCD] determined?

Fig 5D Correct legend: change (E) to (D)

This has been corrected in the revised manuscript.

Referee #3:

In this work, Rambaud et al. discover that the *Drosophila* Ser/Thr kinase Slik plays a crucial role in regulating membrane extension formation, explaining its previously known role in distant cell proliferation. Using a combination of complementary approaches in vitro in S2 cells and in vivo in wing discs, the authors identify a domain in Slik protein that directly shapes the membrane into filopodia, initiating cytoneme formation. This domain does not require Slik kinase activity. Furthermore, membrane elongation can be prevented by phosphorylation of its target Moesin. Finally, they found that the dSTRIPAK complex, the first family of kinases to directly shape the membrane, also plays an essential role in membrane sculpting function by regulating the association of Slik with the plasma membrane.

Although the authors have not yet identified the mechanisms that regulate this membrane-forming induction, the evidence that activity of the C-terminal coiled-coil domain of Slik acts as a novel regulator of the biogenesis of membranes to control the proliferation of cells at a distance is very strong. In addition, the paper contains very interesting findings: unlike other I-BAR domain proteins, Slik^{CCD} binds to Ptdins(4)P and is the first membrane sculpting domain capable of inducing both negative and positive curvatures of Giant Unilamellar Vesicles in vitro. All the data are very well presented and the manuscript is well written. My major problem with this manuscript is why the apical extensions are considered to be cytonemes.

In summary, I feel that this paper merits publication after some revision.

Major points

1) My major concern with this manuscript is if the apical membrane extension are proper

cytonemes. There is no evidence that these membrane protrusions perform signaling in a physiological situation. It would be important to see these apical membrane protrusions by EM.

We agree that EM studies on these protrusions would provide valuable insights; however, this approach is technically demanding. To our knowledge, only one group has successfully visualized cytonemes in wing discs using EM (PMID: 33734293). Unfortunately, we currently lack the necessary resources for comparable studies, and even with EM confirmation, it would remain challenging to confirm the signaling function of the observed apical protrusions.

Most publications on cytonemes implicated in cell-cell signaling involving different pathways and tissue communication in the wing disc have been described at the basolateral and basal side of the disc epithelium. If Slik has general membrane sculpting functions to control biogenesis of various specialized membrane protrusions it would be important to show or at least discuss if Slik could have also sculpting membranes function basally.

Here, the function of Slik in the membrane biogenesis has only been analyzed in the apical plasma membrane of the wing disc. A possible reason for this could be that Slik protein localizes only apically; if this is the case, this has to be well explained in the manuscript.

We previously observed strong apical accumulation of Slik in discs, with the most obvious phenotype (DP-PerM signaling) seemingly linked to effects at the apical membrane. This observation guided our initial focus on the apical plasma membrane. However, to provide a comprehensive view, we have now included additional data in a new supplemental figure (Fig. EV1) showing Slik immunostaining on cross-sections of cryopreserved wing discs to assess its full apical-basal distribution. Consistent with our earlier findings (as noted in the first paragraph of the Results section), we confirmed strong enrichment of Slik, and consequently phospho-Moesin, at the apical side of the DP layer. Interestingly, we also observed a less pronounced but clear enrichment of endogenous Slik and phospho-Moesin at the basal side of the disc (Fig. EV1).

In the new supplemental Figure EV5 A-B, we further demonstrate that upon overexpression of Slik-GFP or Slik^{CCD}-GFP in the wing disc, both proteins also localize to basally located protrusions. Additionally, in the air-sac primordium—a system where cytonemes play a key role in signal exchange—we observed accumulation of Slik^{CCD}-GFP in the basal cytonemes formed by these cells (Figure EV5 C).

Together, these observations suggest that Slik may have a role in sculpting basal membranes. We have added a new paragraph to the Discussion referencing this supplemental data, proposing that Slik could influence basolateral cytoneme formation and function.

2) Although the effect of Slik-GFP overexpression to analyze membrane extensions in the wing disc is shown in Fig 1 and Fig 4 C, experiments of Slit loss of function have been performed in S2 cells but not in the wing disc. I have only found one experiment that has been carried out in the wing disc using Strip dsRNA but in this case SlikGFP is overexpressed simultaneously (Fig 6D). I consider important to show the effect of the absence of Slik in the wing imaginal disc. If confocal

images do not provide enough resolution to visualize the microvilli, EM images could overcome this problem.

Strong depletion or loss of *slik* in discs results in a loss of tissue integrity, with many cells sorting out and undergoing apoptosis, a phenotype resembling and likely related to that of *moesin* mutants. To achieve a more moderate RNAi loss-of-function phenotype, we optimized conditions (temperature, Dicer expression) and, under these conditions, observed an unexpected increase in apical membrane protrusions in Slik-depleted cells (Fig. EV2). Importantly, however, these protrusions did not lead to an increase in PerM cell proliferation (Fig. EV2).

While initially surprising, this result aligns with the antagonistic roles of Slik catalytic and non-catalytic functions in regulating protrusion formation, as we describe in S2 cells in the manuscript (in agreement with prior literature). Specifically, Slik^{CCD} promotes membrane tubulation, while Slik catalytic activity, through Moesin phosphorylation, restricts cytoneme formation. Our findings suggest that in the absence of Slik, reduced active Moesin and actin-membrane contacts increase apical membrane protrusiveness. Consequently, protrusions form, likely under the influence of other mechanisms such as other I-BAR proteins, but these structures lack signaling function. This data has now been incorporated into the Results section, with further discussion in the Discussion section.

3) While Slik seems to promote the delivery of growth signals from DP to PerM cells through an increase in the number and length of apical cytonemes, the signals involved in this transluminal cytoneme-mediated communication have not been identified in this work. Besides, the described apical effect of ectopic DP over the PM could be simply a consequence of the overexpression. The GRAP signal shown in Fig 2 G occurs only under Slik overexpression and not, or barely, in normal conditions.

As we mentioned in response to Reviewer 2's comments, we believe we have identified and are currently investigating the signal involved in transluminal communication. Although this line of research is promising, it is still in progress. Given that this manuscript primarily focuses on Slik's role in membrane shaping and cytoneme formation, we plan to explore the mechanistic details of this pathway in future studies.

Relevant to the reviewer's point, we have identified a gene whose depletion in DP cells produces formation of DP apical cytonemes (*rebuttal Fig 3*) and overproliferation of PerM cells (*rebuttal Fig 4*), similar to the effects observed with Slik overexpression. As these phenotypes are induced both by Slik overexpression and by loss-of-function for this gene, we are confident that the transluminal pro-proliferation signaling described in this manuscript reflects a genuine physiological mechanism rather than an artifact of Slik overexpression. Our data suggest that, under normal conditions, the endogenous pool of the encoded protein restricts DP-PerM cell contact, thereby limiting cytoneme formation. We intend to explore this pathway in greater depth in a separate manuscript.

Figures for reviewers removed.

Minor points:

1) In several places along the manuscript some data are presented as "data are not shown". The data should be shown (at least as supplementary material).

As examples:

"In time-lapse movies, there was also a decrease in cytoneme dynamics in strip/cka-depleted cells (not shown)".

"Although we have obtained experimental evidence indicating that SlikCCD directly binds to actin filaments (data not shown)"....

"We found that SlikCCD binds to Ptdins(4)P rather than Ptdins(4,5)P2 like I-BAR domains do, and SlikCCD sculpts GUVs containing Ptdins(4)P but not Ptdins(4,5)P2 (data not shown)".

We have addressed the instances of "data not shown" throughout the manuscript by either removing the corresponding sentences when they were in the discussion or by including the data.

2) Figure 2 C, E. describe the averaging cytoneme diameter as 172 nm and 181 nm. The authors mention that this size coincides with the averaging diameter of cytonemes found by Tom

Kornberg (Kornberg and Roy, 2014). I find more appropriated to also cite the reference Wood et al., 2021, since this article defines the cytoneme diameter by electron microscopy.

This has been corrected.

3) In Fig 2F, while Slik overexpression increases the length of membrane extensions in S2 cells, its depletion does not result in their shortening; instead, cytonemes appear longer than in control cells. An explanation or a comment on this fact is needed.

This paradoxical effect is now explained in the discussion section: *'The antagonistic roles of Slik^{CCD} and kinase domain could explain why Slik depletion reduces the number of cytonemes but increases their length (Fig 2H). Without Slik, its CCD cannot tubulate the plasma membrane into nascent cytonemes, leading to fewer cytonemes per cell. However, in the absence of Slik kinase activity, Moesin is not activated, failing to regulate cytoneme length, which may be sustained by other I-BAR proteins.'*

4) Fig. 4C. I have not been able to see the promotion of cytoneme formation in DP cells expressing SlikKD-GFP.

The image we presented in Fig 4C was at somewhat lower magnification than 4B, this has been corrected in the revised manuscript.

5) Fig 6F and Fig 6 G are not quoted in the main manuscript text.

This has been corrected.

6) It was a surprise to me to find out that the title of this manuscript was based on the results of the last figure.

The title of the manuscript reflected the overarching discovery of our study. We acknowledge, however, that STRIPAK function is demonstrated in the final figure. Throughout the manuscript, we present a series of experiments that investigate how Slik controls cytoneme biogenesis, highlighting the significance of this process and the critical role of the regulatory network involved. In response to the reviewer's comment, we decided to emphasize Slik's role in cytoneme biogenesis and have updated the title to ***Slik sculpts the plasma membrane into cytonemes to control cell-cell communication.***

Dear Sébastien and David,

Thank you for submitting a revised version of your manuscript. We have now received input from two of the original reviewers, who find that their main concerns have been addressed satisfactorily and recommend acceptance of the manuscript after minor revisions as outlined in their comments. Point 2 by reviewer #3 can be addressed textually.

There also are a few editorial points that need addressing before I can extend official acceptance of the manuscript:

1. Please submit up to five keywords.
2. Please make sure that the order of the sections in the manuscript is as follows: abstract, introduction, results, discussion, materials & methods, data availability section, acknowledgments, disclosure statement and competing interests, references, main figure legends, tables, expanded figure legends.
3. Please check that the funding information is correct and identical both in the manuscript and our online system. Currently, ANR-10-IDEX-0001-02 and the doctoral programmes are missing from our online system.
4. In the Author Checklist file, please make sure that an answer option is selected for all rows in the column D.
5. CRedit has replaced the traditional author contributions section because it offers a systematic, machine-readable author contributions format that allows for more effective research assessment. Please remove the Authors Contributions from the manuscript and use the free text boxes beneath each contributing author's name in our online submission system to add specific details on the author's contribution. More information is available in our guide to authors.
6. Please add a "Disclosure and competing interests statement" section after "Acknowledgments" (further info: <https://www.embopress.org/page/journal/14602075/authorguide#conflictofinterest>).
7. Please update references according to The EMBO Journal style - where there are more than 10 authors on a paper, the first 10 should be listed, followed by 'et al.' Please see further information here: <https://www.embopress.org/page/journal/14602075/authorguide#referencesformat>
8. Please rename the movies into Movie EV1-EV3 and update the callouts accordingly. The legends should be removed from the manuscript text file and zipped with each movie file. Further information is available here: <https://www.embopress.org/page/journal/14602075/authorguide#expandedview>
9. There are references to "data not shown" on page 29, first paragraph. Since our policy does not permit this, and the data appear to be included in specific figure panels, please remove the "data not shown" statement.
10. Our data editors have flagged the following issues in figure legends that need correcting:
 - Please provide figure titles for figures EV 3, EV 4, EV 5.
 - Please define the annotated p values $***$ as well as provide the exact p-values for the same in the legend of figure 5f; as appropriate.
 - Please provide the exact p values in the legends of figures 1c, g; 2f, h; 3a-c; 4b, d-e, h, j; 6a-b, f-g; EV 3.
 - Please note that in figures 4b, d-e, h, j; there is a mismatch between the annotated p values in the figure legend and the annotated p values in the figure file that should be corrected.
 - Please provide information on the number and nature of replicates in the legends of figures 1c, g; 4e; 6f-g; EV 2b; EV 3.
 - Please define the error bars in the legends of figures 1c, g; 2d, f, h; 3a-c; 4b, d-e, j; 5f; 6a-b, f-g; EV 2b; EV 3.
 - Please define the black/yellow arrows in the legend of figure 1c; 4h.
 - Please define the black/white arrowheads in the legend of figure 4g.
11. Papers published in The EMBO Journal are accompanied online by a 'Synopsis' to enhance discoverability of the manuscript. It consists of A) a short (1-2 sentences) summary of the findings and their significance, B) 3-4 bullet points highlighting key results and C) a synopsis image that is 550x300-600 pixels large (width x height, jpeg or png format). You can either show a model or key data in the synopsis image. Please note that the image size is rather small and that text needs to be readable at the final size. Please send us this information together with the revised manuscript.

With best wishes,

Ieva

Ieva Gailite, PhD
Senior Scientific Editor
The EMBO Journal
Meyerhofstrasse 1

We realize that it is difficult to revise to a specific deadline. In the interest of protecting the conceptual advance provided by the work, we recommend a revision within 3 months (17th Mar 2025). Please discuss the revision progress ahead of this time with the editor if you require more time to complete the revisions.

Referee #1:

The manuscript was significantly improved. I have a concern about the nomenclature. "Cytosomes" are not correctly used. The use of cytosome also does not fit the authors' definition. The authors' definition of cytosomes is "Cytosomes are signaling filopodia that facilitate long-range cell-cell communication by forming synapses between cells," as in the abstract. Still, the counting of protrusions does not appear to be performed with the confirmation of the contact of the cytosomes with the neighbor cells.

Furthermore, in the rebuttal letter, the authors wrote that cytosomes and filopodia can not be distinguished. Then, the use of cytosomes should be more carefully used in the manuscript. Most of the observations are on the cellular protrusions and might not be on the cytosomes. Another reviewer also raises this point.

Movie 1 shows stable protrusions but the attached sites are not visible. I think the overall movement of protrusions is small. Control is missing in Movie1

Movie 2 shows the release of some particles at the top of the protrusion with Silk protein. These would be the protrusion-derived or filopodia-derived EVs for cell-cell communication and better if described with citations (PMID 33756122). The tip of cytosomes is cut and engulfed into the contacted cells, but this is essentially the same phenomenon as the release of EVs from the tip of protrusions (PMID 39035026). These points need to be discussed with citations.

In Figure 2 or related, cytosomes can be defined as protrusions with their tips contacting the neighbor cells. If the count is for cellular protrusions, then the label should be "protrusions". The label "cytosomes" is misleading.

Referee #3:

- general summary and opinion about the principal significance of the study, its questions and findings

The manuscript by Rambaud et al. presents significant findings toward understanding the mechanisms underlying the initiation and extension of cytosomes, a process that remains poorly understood. The study identifies the *Drosophila* protein Slik as having membrane-sculpting activity crucial for cytosome biogenesis. Additionally, the dSTRIPAK complex is characterized as a regulator of Slik's association with the plasma membrane. This study is well-conceived and technically impressive, employing multiple approaches to address the research questions. The revised manuscript is notably improved, including new experiments to support their findings. All data are clearly presented and conclusions well-supported.

All my questions and concerns regarding the first revision of this manuscript have been thoroughly addressed. The authors have provided convincing responses. In addition, the authors' answers to the questions raised by other reviewers seem to me to be satisfactory. Although the signaling mechanisms responsible for the proliferative effect induced in the PerM by apical cytosomes overexpressing Slik in the DP remain unidentified and require further investigation, I believe the manuscript is ready for publication in The EMBO Journal without further delay.

- Additional non-essential suggestions for improving the study

1) In Movie 3, cells under Stripak depletion appear to undergo cell death, as suggested by the presence of what seem to be apoptotic bodies. To verify the absence of cytosomes under Stripak mutant conditions, this experiment should be repeated under conditions where apoptosis is inhibited. Alternatively, a higher-quality movie should be provided, which could potentially be achieved by reducing the induction time.

2) I still wonder whether Slik, in addition to its role in the initiation and extension of cytonemes, might also function as a stabilizer of cytonemes based on what it is observed in Fig EV5B,C.

We would like to thank the reviewers for their constructive comments throughout the revision process of our manuscript. We hope these revisions address all remaining concerns.

Best regards,

Sébastien and David

Referee #1:

The manuscript was significantly improved. I have a concern about the nomenclature. "Cytonemes" are not correctly used. The use of cytoneme also does not fit the authors' definition. The authors' definition of cytonemes is "Cytonemes are signaling filopodia that facilitate long-range cell-cell communication by forming synapses between cells," as in the abstract. Still, the counting of protrusions does not appear to be performed with the confirmation of the contact of the cytonemes with the neighbor cells. Furthermore, in the rebuttal letter, the authors wrote that cytonemes and filopodia can not be distinguished. Then, the use of cytonemes should be more carefully used in the manuscript. Most of the observations are on the cellular protrusions and might not be on the cytonemes. Another reviewer also raises this point.

We acknowledge the reviewer's concern regarding the nomenclature of cytonemes. The distinction between cytonemes and filopodia remains an area of active investigation, and while cytonemes are generally considered signaling filopodia, research has yet to establish clear-cut criteria for differentiating them in all contexts. For instance, whether non-signaling filopodia can be converted into cytonemes, or vice versa, remains an open question.

To ensure terminological clarity and maintain consistency with the growing body of literature on specialized protrusions involved in signaling and to maintain ease of readability of the manuscript, we have opted—following further discussion with the editor—to retain the term "**cytonemes**" while also acknowledging the challenges of definitively classifying the structures we observed in S2 cells. This choice allows for a more coherent discussion of our findings within the framework of existing studies.

To address the reviewer's concern, we have revised the text to explicitly state these limitations and clarify that distinguishing cytonemes from filopodia remains difficult in the absence of definitive markers. We have also adjusted the language in relevant sections to reflect that while we refer to these structures as cytonemes, definitive proof of their signaling function in every instance remains to be fully established.

We appreciate the reviewer's feedback, which has helped us refine the manuscript and improve its precision.

Movie 1 shows stable protrusions but the attached sites are not visible. I think the overall movement of protrusions is small. Control is missing in Movie1

Although we assume there are attachment sites for the stable protrusions, this is not something that we have attempted to address in this manuscript. The protrusions are forming in a very narrow space (disc lumen), and as mentioned in the manuscript they extend out to about 4+ μm . Movie 1 shows protrusions forming, extending, and retracting over distances of this scale. We included Movie 1 to show that Slik itself (Slik-GFP) localizes to dynamic protrusions, and this is supported by data in live S2 cells (Fig 2A, Movie 2).

Movie 2 shows the release of some particles at the top of the protrusion with Silk protein. These would be the protrusion-derived or filopodia-derived EVs for cell-cell communication and better if described with citations (PMID 33756122). The tip of cytonemes is cut and engulfed into the contacted cells, but this is essentially the same phenomenon as the release of EVs from the tip of protrusions (PMID 39035026). These points need to be discussed with citations.

We thank the reviewer for pointing out the relevance of extracellular vesicle (EV) literature to our findings. In response, we have revised the discussion to include a detailed consideration of the release of vesicles observed from cytoneme-like protrusions. Specifically, we now describe how these vesicles are reminiscent of extracellular vesicles (EVs) released by I-BAR-induced filopodia, such as MIM, as previously reported (PMID 33756122, PMID 39035026).

In Figure 2 or related, cytonemes can be defined as protrusions with their tips contacting the neighbor cells. If the count is for cellular protrusions, then the label should be "protrusions". The label "cytonemes" is misleading.

We appreciate the reviewer's comment and refer to our previous response, where we acknowledged the challenges in definitively classifying the structures in S2 cells as cytonemes. As discussed, we have clarified these limitations in the manuscript and revised the text to ensure transparency regarding the terminology used.

Referee #3:

- general summary and opinion about the principal significance of the study, its questions and findings

The manuscript by Rambaud et al. presents significant findings toward understanding the mechanisms underlying the initiation and extension of cytonemes, a process that remains poorly understood. The study identifies the *Drosophila* protein Slik as having membrane-sculpting activity crucial for cytoneme biogenesis. Additionally, the dSTRIPAK complex is characterized as a regulator of Slik's association with the plasma membrane. This study is well-conceived and technically impressive, employing multiple approaches to address the research questions. The revised manuscript is notably improved, including new experiments to support their findings. All data are clearly presented and conclusions well-supported.

All my questions and concerns regarding the first revision of this manuscript have been thoroughly addressed. The authors have provided convincing responses. In addition, the authors' answers to the questions raised by other reviewers seem to me to be satisfactory. Although the signaling mechanisms

responsible for the proliferative effect induced in the PerM by apical cytonemes overexpressing Slik in the DP remain unidentified and require further investigation, I believe the manuscript is ready for publication in The EMBO Journal without further delay.

- Additional non-essential suggestions for improving the study

1) In Movie 3, cells under Stripak depletion appear to undergo cell death, as suggested by the presence of what seem to be apoptotic bodies. To verify the absence of cytonemes under Stripak mutant conditions, this experiment should be repeated under conditions where apoptosis is inhibited. Alternatively, a higher-quality movie should be provided, which could potentially be achieved by reducing the induction time.

Although the wing discs exhibit apparent signs of blebbing upon STRIPAK component depletion, which could suggest some cells are undergoing apoptosis, we also observe a loss of cytoneme-like protrusions in S2 cells under similar conditions without the appearance of signs of apoptosis such as nuclear fragmentation (Fig 6C). Indeed, our previous findings demonstrated that STRIPAK component depletion in S2 cells neither induced apparent apoptotic features nor impaired their ability to progress through mitosis (PMID 32960945). This indicates that the blebbing observed in the wing disc and the loss of cytonemes are distinct phenomena and are not necessarily mechanistically connected. We have added this information to the manuscript where we present these results.

2) I still wonder whether Slik, in addition to its role in the initiation and extension of cytonemes, might also function as a stabilizer of cytonemes based on what it is observed in Fig EV5B,C.

We thank the reviewer for this insightful comment. We have now added a discussion point in the manuscript to consider the possibility that Slik not only initiates and extends cytonemes but may also function as a stabilizer.

Dear Sebastien and David,

Thank you for addressing the final editorial issues. I am now pleased to inform you that your manuscript has been accepted for publication.

Before we forward your manuscript to our publishers, I would like to propose some minor edits in the manuscript abstract and the summary section of the synopsis (please see below and the attached file). I have also written a short blurb that will accompany the title of your manuscript in our online table of contents. Please take a look and let me know if any corrections or adjustments are needed:

Blurb:

The Ser/Thr kinase Slik promotes membrane tubulation through its coiled-coil domain and counteracts it via Moesin activation through its kinase domain.

Synopsis:

Cytonemes are specialized cellular protrusions that mediate cell-cell communication at a distance. This study identifies Slik as a unique Ser/Thr kinase that is essential for cytoneme biogenesis, with dual roles in membrane sculpting and regulating cortical stiffness.

If you have any questions, please do not hesitate to contact the Editorial Office. Thank you for this contribution to The EMBO Journal and congratulations on a nice study!

Best wishes,

Ieva
